# Ventilator output splitting interface 'ACRA': Description and evaluation in lung simulators and in an experimental ARDS animal model

Pablo E. Otero[1]*, Lisa Tarragona[1], Andrea S. Zaccagnini[1], Natali Verdier[1], Martin R. Ceballos[1], Emiliano Gogniat[2], Juan M. Cabaleiro[3], Juan D'Adamo[3], Thomas Duriez[3], Pedro Garcia Eijo[3], Guillermo Artana[3]

1 Cátedra de Anestesiología y Algiología, Facultad de Ciencias Veterinarias, Buenos Aires, Argentina,
2 Capítulo de Kinesiología Intensivista, Sociedad Argentina de Terapia Intensiva, Buenos Aires, Argentina,
3 Laboratorio de Fluidodinámica, Facultad de Ingeniería, Universidad de Buenos Aires—CONICET, Buenos Aires, Argentina

* potero@fvet.uba.ar

## Abstract

The current COVID-19 pandemic has led the world to an unprecedented global shortage of ventilators, and its sharing has been proposed as an alternative to meet the surge. This study outlines the performance of a preformed novel interface called 'ACRA', designed to split ventilator outflow into two breathing systems. The 'ACRA' interface was built using medical use approved components. It consists of four unidirectional valves, two adjustable flow-restrictor valves placed on the inspiratory limbs of each unit, and one adjustable PEEP valve placed on the expiratory limb of the unit that would require a greater PEEP. The interface was interposed between a ventilator and two lung units (phase I), two breathing simulators (phase II) and two live pigs with heterogeneous lung conditions (phase III). The interface and ventilator adjustments tested the ability to regulate individual pressures and the resulting tidal volumes. Data were analyzed using Friedman and Wilcoxon tests test ($p < 0.05$). Ventilator outflow splitting, independent pressure adjustments and individual tidal volume monitoring were feasible in all phases. In all experimental measurements, dual ventilation allowed for individual and tight adjustments of the pressure, and thus volume delivered to each paired lung unit without affecting the other unit's ventilation—all the modifications performed on the ventilator equally affected both paired lung units. Although only suggested during a dire crisis, this experiment supports dual ventilation as an alternative worth to be considered.

## Introduction

The current COVID-19 pandemic has led to an unprecedented global demand of mechanical ventilators and shortage of such equipment could have substantially increased the mortality rate of the disease [1]. The world ran a race to scale up production of ventilators, but this process was not fast enough, and ventilators, together with human resources and other medical

**Data Availability Statement:** The data repository is now here: https://github.com/ACRA2020/Pinch_Valve_design.

**Funding:** This study was funded by the University of Buenos Aires 'UBATEC', 'Programa RespirAR' The funders had no role in study design, data collection and analysis, decision to publish, or preparation of the manuscript.

**Competing interests:** The authors have declared that no competing interests exist.

supplies remained the bottleneck [2]. This has left the world, especially low and middle-income countries, facing a prospective catastrophic scenario where medical workers continue to be forced to ration these goods through the toughest triage [1, 3]. Sharing of a ventilator by multiple patients was originally proposed by Sommer et al. [4] to meet disaster surge and is currently being considered as an alternative for the COVID-19 crisis. However, this strategy has been debated. Although several medical societies have recently developed a joint statement [5] and advised against it addressing several shortcomings, the United States' Food & Drug Administration (FDA) has authorized the emergency use of continuous ventilator splitters exclusively due to the current COVID-19 outbreak [6]. In light of the current situation, several studies that support alternatives for dual ventilation have become rapidly available to the scientific community [7–9]. The favorable results most of them have obtained encouraged us to build a preformed interface designed to split ventilator outflow into two breathing systems. Based on the method of flow restriction using adjustable valves and under pressure control ventilation mode, it would permit individual and tight adjustments on the driving pressure [ΔP; plateau pressure (Ppl)—positive end-expiratory pressure (PEEP)] of each paired unit. Its distinct features are that it is assembled using components that were previously approved by medical regulatory agencies [i.e., FDA, ANMAT] and that it incorporates two analog manometers for the specific and individual monitoring of the working pressure [i.e. peak inspiratory pressure (PIP), Ppl, PEEP].

This paper outlines the functioning of the novel ventilator output splitting interface called 'ACRA' (acronym for Enhanced Capacity of Mechanical Ventilators, in Spanish) and tests its performance when interposed between a standard ICU mechanical ventilator (Nellcor Puritan Bennet 760 Ventilator; Covidien, Mansfield, MA) and two lung units (ACCU LUNG Precision Test Lung; phase I), two breathing simulators (ASL 5000, InGMAR Medical, Pittsburgh, PA; phase II) and two live pigs with heterogeneous lung conditions (phase III).

## Materials and methods

The ACRA interface was designed to be placed between a mechanical ventilator and two standard breathing systems. With the use of four unidirectional valves, one adjustable PEEP valve placed in-line and two pinch valves designed ad hoc, PIP and PEEP of each lung unit can be manually and individually controlled. The inspiratory pressure of each unit is controlled by introducing a pressure drop through the manual adjustment of the pinch valve located in each patient's inspiratory circuit. The PIP achieved in each unit is, therefore, equal or lower than the one imposed by the ventilator. The adjustable PEEP valve, which had a previously mounted 3D-printed collar to permit its use in line [10], is placed on the expiratory limb of the lung unit that requires a greater PEEP than that set on the ventilator. Therefore, the PEEP on the unit without the adjustable PEEP valve corresponds to the PEEP set on the ventilator, whereas the PEEP of the unit with the adjustable PEEP valve equals to the sum of the PEEP value set on the ventilator and the value set on the adjustable PEEP valve. Two analog manometers incorporated to the interface sense pressure downstream of each pinch valve and allow for the measurement of the ΔP of each unit. Unidirectional valves inhibit the flow in undesired directions in both the inspiratory and expiratory phase. A 15 cm long and 6-mm bore bypass circuit between the expiratory and inspiratory limbs bypasses flow parallel to the two paired units in order to ensure predictable behavior of the ventilator [11].

The interface was constructed with disposable materials approved for medical use, including the standard silicone tubing of the pinch valves. These valves were designed to completely isolate fresh gas flow from internal valve parts.

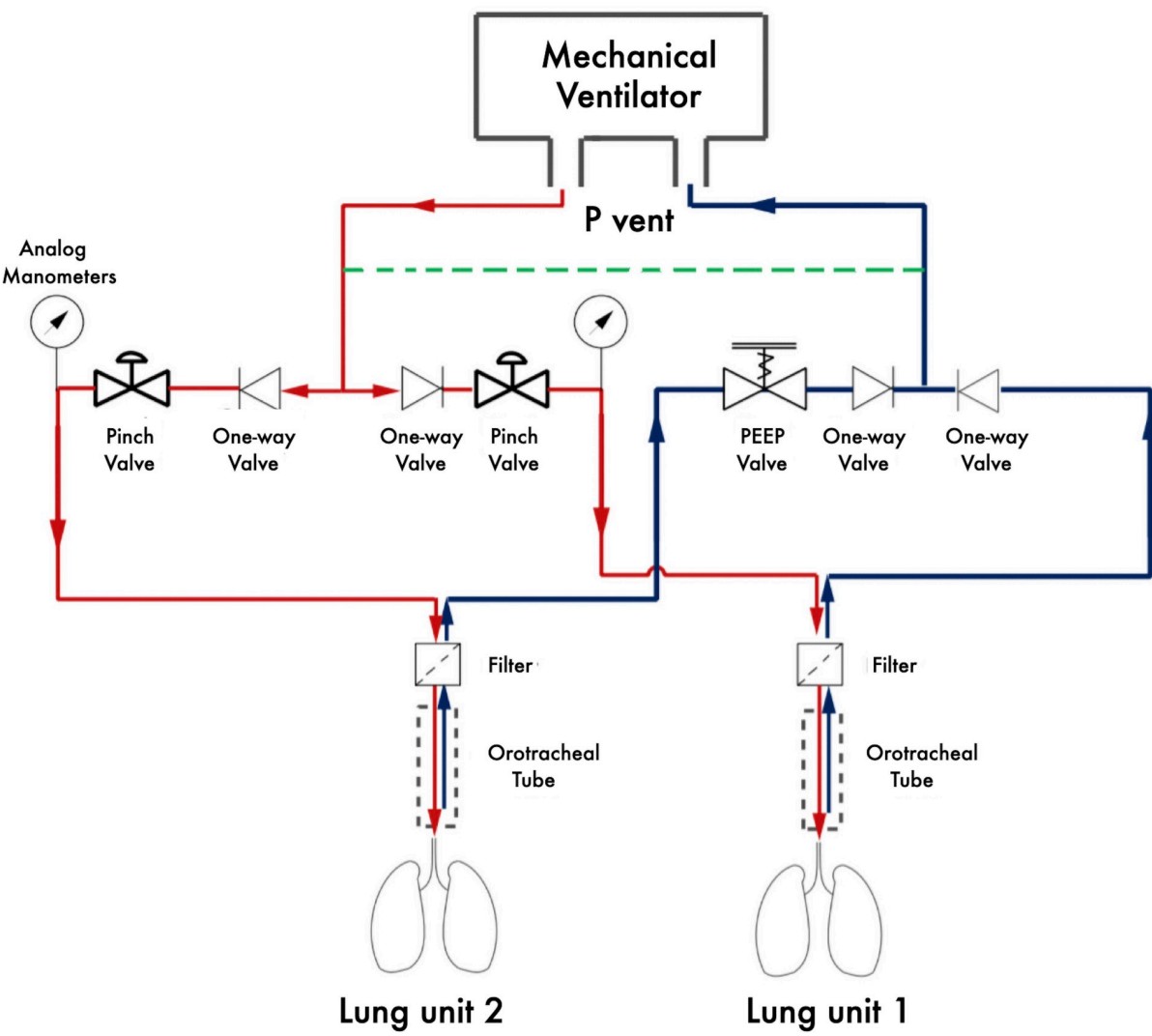

**Fig 1. One-line diagram of the ACRA interface.** Inspiration and expiration circuits are colored in red and black, respectively. A dashed green line represents a bypass circuit of small diameter.

An illustrative one-line diagram of the ACRA interface is presented in Fig 1 and renders of the prototype are shown in Fig 2. An illustrative video showing how to assemble the system is given as an online supplement.

The study design was outlined in three phases. The ACRA was tested using lung units, breathing simulators and live animal porcine models in phase I, II and III, respectively. In all of the phases, the ACRA was attached to the inspiratory and expiratory ports of a standard ICU mechanical ventilator through standard corrugated tubing, and to the two corresponding lung units through standard breathing circuits. HMEF and HEPA filters were interposed accordingly. Standard operational self-tests were performed prior to each experiment and compliance of the whole system was calculated.

Mechanical ventilation was performed using pressure control mode. Data of pressure and volume of each lung unit were monitored individually. Data obtained from the ventilator and analog manometers were compared with data obtained from each unit's respiratory mechanics

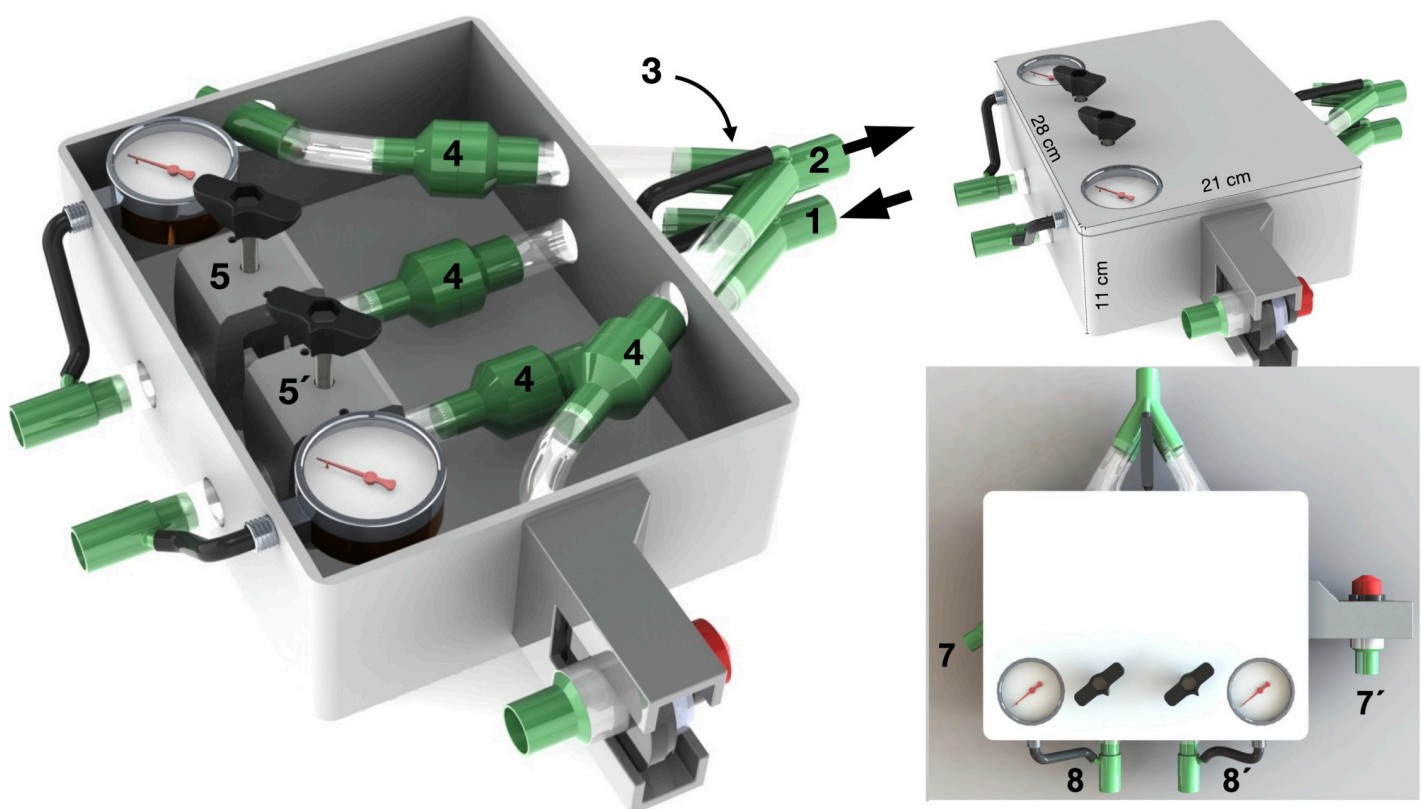

**Fig 2. Three-dimensional volume renderings of the ACRA interface.** (1) Connection for inspiratory limb from ventilator, (2) connection for expiratory limb to ventilator, (3) bypass circuit, (4) unidirectional valves, (5) and (5') pinch valves, (6) adjustable PEEP valve, (7) and (7') connection for expiratory tubes from paired units, (8) and (8') connection for inspiratory tubes to paired units.

monitor (FluxMed®, MBMed, Buenos Aires, Argentina) in phases I and III, and data obtained from the breathing simulators (Ing Mar ASL 500) in phase II.

## Phase I

In order to determine if the interposing of the ACRA would guarantee independent delivery of pressure and tidal volume to two paired lung units with similar compliance and resistance, manual adjustments of the pinch valves of each unit and of the adjustable PEEP valve on the ACRA, together with modifications on the PIP and PEEP set on the ventilator were performed. Adjustments consisted of sequential increments or decrements of pressure on each variable, both on the ventilator and on the ACRA device. Measurements were repeated three times and data was averaged for further analysis.

Additionally, variations of VTe were recorded in lung unit 1 after performing a sole modification of the respiratory rate on the ventilator using a fixed or variable inspiratory time. Each variation of the respiratory rate was performed while applying a flow restriction of 0, 10 and 20 cm $H_2O$ in lung unit 1, which resulted in a PIP of 35, 25 and 15 cm $H_2O$, respectively.

## Phase II

The capacity of the ACRA to guarantee the provision of a $V_{Te}$ of 6 mL/kg (i.e. 420 mL) to two paired breathing simulators with heterogeneous conditions was evaluated. Compliance and

resistance of simulator 1 was set to represent a fixed mild ARDS condition whereas simulator 2 was set to represent a progressively worsening ARDS.

In order to measure $P_{pl}$ and to determine the presence of intrinsic PEEP in both simulators, inspiratory and expiratory breath holds were applied, respectively. In order to avoid back flow during these maneuvers, the bypass circuit was clamped. These values obtained on the analog manometers were compared with the data obtained from the simulator. The $V_{Te}$ of each paired simulator was determined through a five-seconds occlusion on the endotracheal tube of the paired unit, deducting the fall from the global $V_{Te}$ previously read on the ventilator. This value was compared with the data obtained from the individual flow sensors of the breathing simulators. Inspiratory efforts and a brief disconnection (i.e., four respiratory cycles) were simulated in simulator 2 and the resulting modifications in simulator 1 were recorded.

## Phase III

The capacity of the ACRA to guarantee the provision of an independent lung protective ventilation strategy to two paired pig models with heterogeneous conditions was evaluated.

This experiment was performed after obtaining Ethical Committee's Approval of the University of Buenos Aires (IACUC 2020/08).

A total of four Landrace pigs weighing 18.5 ± 1.3 kg were included and allocated in pairs. This phase consisted in two steps, and this experiment was repeated twice.

**Animal model.**  Pigs were placed in the supine position and anesthetized using standard total intravenous anesthesia techniques, which included the constant rate administration of atracurium (0.7 mg/kg/h). After orotracheal intubation using 7 mm ID cuffed endotracheal tube, each animal was connected to a single mechanical ventilator. Additional instrumentation consisted of peripheral and central venous accesses, and a peripheral arterial access. Heart rate, respiratory rate, percentage of hemoglobin saturated with oxygen ($SpO_2$), end-tidal carbon dioxide, invasive arterial blood pressure, oesophageal temperature, perfusion index and plethysmographic variability index, inspired ($FiO_2$) and expired fractions of oxygen, and respiratory mechanics were continuously monitored in all of the animals. Arterial blood samples were obtained and analyzed using portable analyzer (Osmetech OPTI CCA, Osmetech, Inc. Roswell, GA, USA). Additionally, stroke volume was estimated using transthoracic ultrasonography in all of the animals. Lactated Ringer's solution was initially administered to all animals at a rate of approximately 5 mL/kg/h and adjusted according to PVI. Intravenous infusion of norepinephrine (initial dose 0.01 μg/kg/min) was administered to maintain arterial blood pressure, stroke volume and PVI within normal limits.

**Step 1: Dual ventilation set up.**  The ventilator was set to PIP of 35 cm $H_2O$, a PEEP of 5 cm $H_2O$, I:E ratio of 1:2 $FiO_2$ of 0.4 and $f$ of 12 breaths/min. Initially, both pinch valves were closed and the adjustable PEEP valve opened. Immediately after animal 1 was connected to its corresponding breathing circuit, its corresponding pinch valve was manually opened to generate the necessary driving pressure to obtain a $V_{Te}$ of 10 mL/kg ($V_{Te1}$). Subsequently, animal 2 was connected to its corresponding breathing circuit and its pinch valve opened to generate the necessary driving pressure to obtain a $V_{Te}$ of 10 mL/kg ($V_{Te2}$), which was calculated from the summed value displayed on the ventilator screen ($V_{Te2}$ = Global $V_{Te}$—$V_{Te1}$) and corroborated with the data obtained from the respiratory mechanics monitor's flow sensors connected to each animal.

After a 20-minute stabilization period, data from each animal were recorded and these considered baseline values.

**Step 2: Dual ventilation in heterogeneous models.**  In order to reproduce an experimental ARDS model in one pig of the pair, lung lavages with normal saline (30 mL/kg at 37˚C)

were performed. Guided by the evaluation of lung aeration and $SpO_2$, lavages were repeated until $PaO_2/FiO_2 < 200$ mm Hg at PEEP 10 cm $H_2O$, as previously reported by Tusman et al. [12]. Both paired units were continuously ventilated in dual mode for eight hours. Periodic adjustments of the pinch and PEEP valves of the ACRA as well as variations in ventilator settings (i.e. $V_{Te}$, $f$, I:E, $\Delta P$) were performed in order to maintain $PaO_2$ above 60 mmHg, partial pressure of arterial carbon dioxide below 60 mmHg and pH above 7.20 in both paired pigs. In the face of eventual $V_{Te}$ variations (± 10 mL) or the presence of intrinsic PEEP, attempts to return to protective ventilation and normal gas exchange values through the adjustments of the ventilator and/or ACRA settings were performed.

## Statistical analysis

Friedman, Wilcoxon and Dunn's *post hoc* for multiple comparisons tests were used to determine statistical significance of the data obtained in phase I. Wilcoxon matched-pairs signed rank test was used to determine statistical significance of the data obtained in phase II. Significance was considered when $p < 0.05$. Data obtained from the breathing simulators' flow sensors was compared to the pressure (i.e. PIP, PEEP and Ppl) and volume (i.e. $V_{Te}$) values obtained from the analog manometers and ventilator, respectively.

## Results

The ACRA was correctly assembled between the mechanical ventilator and all of the paired units in all of the phases. Operational self-tests were successful and there were no leaks detected in any of the phases. Average system compliance in the three phases was 4.7 ± 0.2 mL/cm $H_2O$.

## Phase I

The ACRA permitted successful splitting of the output of the ventilator, and the manual adjustment of its valves allowed for individual and tight adjustments of the pressure, and thus volume delivered to each paired lung unit, without affecting the other unit's ventilation. Manual control of the adjustable PEEP valve on the ACRA permitted the set of the PEEP of the corresponding lung unit, without affecting the PEEP of the pair.

The interposing of the ACRA between the ventilator and the paired lung units did not alter the normal functioning of the ventilator, and all the modifications performed on the ventilator equally affected both paired lung units.

The results obtained from all variations tested are presented in Tables 1 and 2.

Variations of VTe in lung unit 1 during respiratory rate modifications are presented in Tables 3 and 4.

## Phase II

The interposing of the ACRA was successful to provide an independent $V_{Te}$ and PEEP to each of the paired breathing simulators with different lung mechanics. Inspiratory and expiratory breath holds maneuvers permitted the measurement of the $P_{pl}$ and intrinsic PEEP, respectively, of each paired simulator. A five-seconds occlusion of the endotracheal tube of one simulator allowed for an accurate measurement of the $V_{Te}$ of the paired unit, which was corroborated with the data obtained from the simulator (Table 5). Inspiratory efforts simulated on simulator 2 generated trigger ventilations that affected simulator 1, but a brief disconnection of the simulator 2 did not affect ventilation of the pair.

**Table 1. Proposed variations on the peak inspiratory pressure and positive end-expiratory pressure of the ventilator and the resulting values of peak inspiratory pressure, positive end-expiratory pressure, driving pressure and expired tidal volume on each lung unit.**

| | | Lung unit 1 | | | | Lung unit 2 | | | |
|---|---|---|---|---|---|---|---|---|---|
| | | PIP | PEEP | $\Delta P$ | $V_{Te}$ | PIP | PEEP | $\Delta P$ | $V_{Te}$ |
| | | cm $H_2O$ | cm $H_2O$ | cm $H_2O$ | Mean (SD) | cm $H_2O$ | cm $H_2O$ | cm $H_2O$ | Mean (SD) |
| | | | | | mL | | | | mL |
| Ventilator setting | Initial setting | 20 | 5 | 15 | 518.3 (4.2) | 20 | 6 | 14 | 450.0 (16.6) |
| | Intervention | | | | | | | | |
| | • PIP: increased 5 cm $H_2O$ <br> • PEEP: increased 5 cm $H_2O$ | 25 | 10 | 15 | 535.3 (15.9) | 25 | 11 | 14 | 482.0 (10.0) |
| | • PIP: increased 7 cm $H_2O$ <br> • PEEP: increased 8 cm $H_2O$ | 32 | 18 | 14 | 145.0 (17.7) | 32 | 19 | 13 | 124.3 (17.2) |
| | • PIP: increased 10 cm $H_2O$ <br> • PEEP: increased 7 cm $H_2O$ | 42 | 25 | 17 | 111.7 (10.8) | 42 | 26 | 16 | 123.3 (13.8) |
| | • PIP: decreased 9 cm $H_2O$ <br> • PEEP: decreased 9 cm $H_2O$ | 33 | 16 | 17 | 145.7 (11.9) | 33 | 17 | 16 | 141.7 (17.5) |
| | • PIP: decreased 10 cm $H_2O$ <br> • PEEP: increased 4 cm $H_2O$ | 23 | 12 | 11 | 372.0 (12.8) | 24 | 13 | 11 | 415.0 (8.9) |
| | • PIP: increased 5 cm $H_2O$ <br> • PEEP: increased 5 cm $H_2O$ | 19 | 4 | 15 | 434.0 (13.5) | 19 | 5 | 14 | 437.7 (16.0) |

PEEP = positive end-expiratory pressure, PIP = peak inspiratory pressure, $V_{Te}$ = expired tidal volume, $\Delta P$ = driving pressure.

Initial PIP and PEEP values on the ventilator were set to 20 and 5 cm $H_2O$, respectively.

Statistically significant differences between the values of pressure and volume obtained from the analog manometers and ventilator, respectively, and those observed on the simulator software were not observed ($p < 0.05$; Table 6).

## Phase III

The ACRA allowed for independent ventilation of two paired pigs with different lung mechanics. The interposing of the ACRA did not preclude lung protective ventilation and permitted normalization of lung condition and gas exchange in this model of ARDS (Table 7). In the proposed ARDS scenario ($PaO_2/FiO_2 < 200$ mm Hg at PEEP 10 cm $H_2O$), the required adjustment on the ventilator setting (e.g. increased respiratory rate) imposed changes (e.g. adjust a new driving pressure to reduced $V_{Te}$ to maintain minute ventilation) on the other model. The analog manometers of the ACRA allowed a continuous and individualized monitoring of PIP and PEEP of each paired animal. Alternatively, through the use of the inspiratory and expiratory breath holds, $P_{pl}$ and intrinsic PEEP were also recorded. Measurement of both unit's $V_{Te}$ was feasible and its value was compared with the data obtained from the respiratory mechanics monitor.

## Discussion

This study presents and evaluates the performance of a preformed novel interface specifically designed for the purpose of dual ventilation. It permits the splitting of the ventilator output between two paired lung units. Used under pressure control ventilation mode, this interface allowed for selective and tight titration of PIP, PEEP and therefore $\Delta P$ and tidal volumes in both units and in all of the proposed clinical scenarios, without affecting neither the functioning of the ventilator nor the stability of the paired unit. Additionally, this novel interface was

**Table 2. Proposed variations of pressure on the pinch valves of both lung units (via adjustment of the flow restriction) and the positive end-expiratory pressure adjustable valve and the resulting values of peak inspiratory pressure, positive end-expiratory pressure, driving pressure and expired tidal volume on each lung unit.**

| | | Lung unit 1 | | | | Lung unit 2 | | | |
|---|---|---|---|---|---|---|---|---|---|
| | | PIP | PEEP | ΔP | $V_{Te}$ | PIP | PEEP | ΔP | $V_{Te}$ |
| | | cm $H_2O$ | cm $H_2O$ | cm $H_2O$ | Mean (SD) | cm $H_2O$ | cm $H_2O$ | cm $H_2O$ | Mean (SD) |
| | | | | | mL | | | | mL |
| **ACRA interface adjustments** | **Initial setting** | 30 | 5 | 25 | 808.0 (20.4) | 30 | 6 | 24 | 775.7 (7.6) § |
| | **Intervention** | | | | | | | | |
| | • $PV_1$: increased restriction 5 cm $H_2O$ <br>• $PV_2$: no action <br>• $PEEP_2$: no action | **25** | 5 | 20 | 652.3 (17.2) | 30 | 6 | 24 | 776.3 (17.8) § |
| | • $PV_1$: increased restriction 10 cm $H_2O$ <br>• $PV_2$: no action <br>• $PEEP_2$: no action | **15** | 5 | 10 | 279.3 (22.5) † | 30 | 6 | 24977 | 774.3 (10.0) § |
| | • $PV_1$: no action <br>• $PV_2$: increased restriction 15 cm $H_2O$ <br>• $PEEP_2$: not action | 15 | 5 | 10 | 266.0 (20.7) † | **15** | 6 | 10 | 273.7 (9.5) |
| | • $PV_1$: no action <br>• $PV_2$: decreased restriction 10 cm $H_2O$ <br>• $PEEP_2$: increased 2 cm $H_2O$ | 15 | 5 | 10 | 258.0 (22.7) † | **25** | **8** | 17 | 514.7 (13.5) ß |
| | • $PV_1$: decreased restriction 10 cm $H_2O$ <br>• $PV_2$: no action <br>• $PEEP_2$: no action | **25** | 5 | 20 | 674.7 (8.1) μ | 25 | 8 | 17 | 525.3 (6.1) ß |
| | • $PV_1$: no action <br>• $PV_2$: decrease restriction 3 cm $H_2O$ <br>• $PEEP_2$: increased 2 cm $H_2O$ | 25 | 5 | 20 | 684.3 (13.6) μ | **28** | **10** | 18 | 544.0 (12.1) ¥ |
| | • $PV_1$: increased restriction 8 cm $H_2O$ <br>• $PV_2$: no action <br>• $PEEP_2$: no action | **17** | 5 | 12 | 356.0 (10.8) | 28 | 10 | 18 | 525.0 (8.9) ¥ |
| | • $PV_1$: decreased restriction 8 cm $H_2O$ <br>• $PV_2$: decreased restriction 3 cm $H_2O$ <br>• $PEEP_1$: increased 5 $H_2O$ <br>• $PEEP_2$: full open | **25** | **10** | 0 | 556.7 (10.1) Ω | **25** | 11 | 14 | 488.7 (9.1) ∞ |
| | • $PV_1$: no action <br>• $PV_2$: full restriction <br>• $PEEP_1$: no action <br>• $PEEP_2$: no action | 25 | 10 | 0 | 528.7(9.5) Ω | **0** | 11 | 0 | 0 |
| | • $PV_1$: full restriction <br>• $PV_2$: decreased restriction 25 cm $H_2O$ <br>• $PEEP_1$: no action <br>• $PEEP_2$: no action | **0** | 10 | 0 | 0 | **25** | 11 | 0 | 491.3 (10.0) ∞ |

PEEP = positive end-expiratory pressure, PIP = peak inspiratory pressure, $V_{Te}$ = expired tidal volume, ΔP = driving pressure. $PV_1$ and $PV_2$: pinch valve 1 and 2, respectively.

†: p = 0.53

μ: p = 0.75

Ω: p = 0.25

§: p > 0.99

ß: p = 0.5

¥: 0.25, ∞: p = 0.25.

Initial PIP and PEEP values on the ventilator were set to 30 and 5 cm $H_2O$, respectively.

**Table 3. Resulting variations of VT$_e$ (mL) in paired lung units after a sole modification of the respiratory rate in the ventilator using a fixed inspiratory time of 2 seconds, while applying a flow restriction of 0, 10 and 20 cm H$_2$O in lung unit 1, which resulted in a PIP of 35, 25 and 15 cm H$_2$O, respectively.**

| | Lung unit 1 | | | | Lung unit 2 | | | |
|---|---|---|---|---|---|---|---|---|
| Intervention | PIP | PEEP | V$_{Te}$ | *p* value | PIP | PEEP | V$_{Te}$ | *p* value |
| | cm H$_2$O | cm H$_2$O | Mean (SD) | | cm H$_2$O | cm H$_2$O | Mean (SD) | |
| | | | mL | | | | mL | |
| • RR: 10 (Ti: 2.0 s) | 35 | 3 | 770.7 (9.0) | 0.19 | 35 | 16 | 186.0 (5.3) | 0.19 |
| • RR: 15 (Ti: 2.0 s) | 35 | 3 | 773.3 (3.1) | | 35 | 16 | 185.3 (5,5) | |
| • RR: 20 (Ti: 2.0 s) | 35 | 3 | 658.0 (7.8) | | 35 | 16 | 145,3 2,1 | |
| • RR: 10 (Ti: 2.0 s) | 25 | 3 | 647.7 (4.7) | 0.19 | 35 | 16 | 179.3 (6.0) | 0.94 |
| • RR: 15 (Ti: 2.0 s) | 25 | 3 | 640.3 (6.1) | | 35 | 16 | 181.7 (1.5) | |
| • RR: 20 (Ti: 2.0 s) | 25 | 3 | 635.0 (7.0) | | 35 | 16 | 176.3 (23.4) | |
| • RR: 10 (Ti: 2.0 s) | 15 | 3 | 271.3 (2.5) | 0.19 | 35 | 16 | 185.0 (5.6) | 0.055 |
| • RR: 15 (Ti: 2.0 s) | 15 | 3 | 255.0 (2.0) | | 35 | 16 | 182.0 (2.0) | |
| • RR: 20 (Ti: 2.0 s) | 15 | 3 | 270.7 (4.0) | | 35 | 16 | 175.3 (3.1) | |

PEEP = positive end-expiratory pressure, PIP = peak inspiratory pressure, V$_{Te}$ = expired tidal volume, RR = respiratory rate, Ti = inspiratory time, s = second.

tested first in lung units, then in sophisticated breathing simulators, and finally in live animal experimental models.

Mechanical ventilation of multiple units has been previously proposed by Sommer et al. [4] and Paladino et al. [13], who suggested that this technique should only be applied in crisis situations with shortage of equipment. Several opposing views to this technique have been stated advocating the inability to deliver different working pressures to the paired lungs, the impossibility to achieve a required tidal volume to each patient, the risks of cross-infection, the difficulties in monitoring both patients simultaneously, the difficulties arising from one patient deteriorating suddenly or having a cardiac arrest and finally, ethical issues [5, 14–16]. However, during the actual COVID-19 pandemic and due to the unprecedented and enormous worldwide requirement of available hospital resources, an increasing number of publications presenting novel ideas, suggestions and guidelines of use of this technique overcoming the

**Table 4. Resulting variations of VT$_e$ (mL) in paired lung units after a sole modification of the respiratory rate in the ventilator using a variable inspiratory time of 2 seconds, while applying a flow restriction of 0, 10 and 20 cm H$_2$O in lung unit 1, which resulted in a PIP of 35, 25 and 15 cm H$_2$O, respectively.**

| | Lung unit 1 | | | | Lung unit 2 | | | |
|---|---|---|---|---|---|---|---|---|
| Intervention | PIP | PEEP | V$_{Te}$ | *p* value | PIP | PEEP | V$_{Te}$ | *p* value |
| | cm H$_2$O | cm H$_2$O | Mean (SD) | | cm H$_2$O | cm H$_2$O | Mean (SD) | |
| | | | mL | | | | mL | |
| • RR: 10 (Ti: 2.0 s) | 35 | 3 | 760.3 (1.5) | 0.27 | 35 | 16 | 171.7 (3.2) | 0.39 |
| • RR: 15 (Ti: 1.3 s) | 35 | 3 | 754.7 (4.7) | | 35 | 16 | 172.0 (4.6) | |
| • RR: 20 (Ti: 1.0 s) | 35 | 3 | 754.3 (5.0) | | 35 | 16 | 174.0 (2.6) | |
| • RR: 10 (Ti: 2.0 s) | 25 | 3 | 707.3 (3.1) A | (A-B) 0.66 | 35 | 16 | 160.0 (7.0) | 0.39 |
| • RR: 15 (Ti: 1.3 s) | 25 | 3 | 510.0 (7.2) B | (A-C) 0.04† | 35 | 16 | 152.3 (1.5) | |
| • RR: 20 (Ti: 1.0 s) | 25 | 3 | 395.0 (17.3) C | (B-C) 0.66 | 35 | 16 | 153.3 (4.0) | |
| • RR: 10 (Ti: 2.0 s) | 15 | 3 | 447.7 (9.5) A | (A-B) 0.66 | 35 | 16 | 170.0 (4.4) | 0.19 |
| • RR: 15 (Ti: 1.3 s) | 15 | 3 | 311.0 (3.0) B | (A-C) 0.04† | 35 | 16 | 161.0 (4.6) | |
| • RR: 20 (Ti: 1.0 s) | 15 | 3 | 242.0 (4.6) C | (B-C) 0.66 | 35 | 16 | 158.7 (3.2) | |

PEEP = positive end-expiratory pressure, PIP = peak inspiratory pressure, V$_{Te}$ = expired tidal volume, RR = respiratory rate, Ti = inspiratory time, s = second.

† statistically significant.

**Table 5. Simulation of progressively worsening acute respiratory distress syndrome conditions on breathing simulator 2 during dual ventilation using the ACRA device.**

| Simulated ARDS conditions on SIM2 | | Ventilator | | SIM1 | | | | | SIM2 | | | | |
|---|---|---|---|---|---|---|---|---|---|---|---|---|---|
| Crs | Rrs | PIP | PEEP | PIP | PEEP | Ppl | ΔP | VTe | PIP2 | PEEP | Ppl | ΔP | VTe |
| (mL/cm H2O) | (cm H2O/L/s) | (cm H2O) | (cm H2O) | (cm H2O) | without added PEEP (cm H2O) | (cm H2O) | (cm H2O) | (mL) | (cm H2O) | with added PEEP (cm H2O) | (cm H2O) | (cm H2O) | (mL) |
| 45 | 3 | 35 | 5 | 14.8 | 5.3 | 13.6 | 8.3 | 429 | 14.0 | 5.0 | 13.8 | 8.8 | 406 |
| 30 | 10 | 35 | 5 | 15.0 | 5.1 | 13.6 | 8.5 | 425 | 33.0 | 16.0 | 32.8 | 16.8 | 433 |
| 20 | 15 | 35 | 5 | 14.8 | 4.9 | 13.2 | 8.3 | 430 | 34.6 | 11.0 | 34.5 | 23.5 | 420 |
| 10 | 20 | 35 | 5 | 14.8 | 4.9 | 13.4 | 8.5 | 430 | 40.7 | 10.6 | 40.4 | 29.8 | 323 |
| 10 | 20 | 40 | 5 | 15.0 | 4.9 | 14.3 | 9.4 | 462 | 46.0 | 10.2 | 46.0 | 35.8 | 385 |

ARDS = acute respiratory distress syndrome, Crs = respiratory system compliance, PEEP = positive end-expiratory pressure, PIP = peak inspiratory pressure, Ppl = plateau pressure, Rrs = respiratory system resistance, SIM1 = simulator 1, SIM2 = simulator 2, VTe = expired tidal volume, ΔP = driving pressure. Resulting variations on the peak inspiratory pressure, positive end-expiratory pressure, plateau pressure, driving pressure and expired tidal volume on both paired simulators. Each simulated condition was performed under fixed ventilator settings (respiratory rate = 20; inspiratory:expiratory ratio = 1:2; FIO2 = 0.5; peak inspiratory pressure = 35 to 40 cm $H_2O$; positive end-expiratory pressure = 5 cm $H_2O$) and fixed mild acute respiratory distress syndrome condition on simulator 1 (respiratory system compliance = 50 mL/cm $H_2O$, respiratory system resistance = 3 cm $H_2O$/L/s).

aforementioned limitations are currently being released [7–9, 17–19]. In our study, we outlined the design and evaluated the functioning of a preformed novel interface with the aim to enrich the available proposals and to provide a solution for the current lack of resources in an uncertain evolution of the COVI-19 pandemic. The main feature of this interface is that, conversely to the so far presented alternatives, this is the first preformed unit that provides continuous and analogue monitoring of the main controlled variable (i.e., pressure), is economical, easily reproducible, and minimizes the risk of accidental incorrect assembly.

The ACRA presented in this study was based on restriction of flow in the inspiratory circuit through the fine turning of the knob of a pinch valve. This method allowed real-time and tight manual adjustments of the inspiratory flow, and is the same principle currently suggested by several authors [7–9, 17]. However, it must be considered that, in pressure-controlled

**Table 6. Comparison of the values of peak inspiratory, positive end-expiratory and plateau pressures, and global and individual expired tidal volumes recorded from the analog manometers and ventilator, respectively, with those values obtained from both simulators' software during the five proposed acute respiratory distress syndrome conditions simulated on simulator 2.**

| | Analog manometer values | Simulator values | p-value |
|---|---|---|---|
| PIP SIM1 (cm $H_2O$) | 15 (15–15) | 14.8 (14.8–15) | 0.125 |
| PIP SIM2 (cm $H_2O$) | 34 (14–44) | 34.6 (14–46) | 0.250 |
| PEEP SIM1 (cm $H_2O$) | 5 (5–5) | 4.9 (4.9–5.3) | >0.999 |
| PEEP SIM2 (cm $H_2O$) | 10 (5–15) | 10.6 (5–16) | 0.125 |
| Ppl SIM1 (cm $H_2O$) | 16 (13–17) | 13.7 (13.4–14.9) | 0.156 |
| Ppl SIM2 (cm $H_2O$) | 31(14–43) | 34.5 (13.8–46) | 0.125 |
| | Ventilator values | Simulator values | p-value |
| $V_{Te}$ SIM1 (mL) | 411 (401–484) | 429 (425–462) | 0.218 |
| $V_{Te}$ SIM2 (mL) | 402 (344–454) | 406 (323–433) | 0.375 |

PEEP = positive end-expiratory pressure, PIP = peak inspiratory pressure, Ppl = plateau pressure, SIM1 = simulator 1, SIM2 = simulator 2, VTe = expired tidal volume.

Statistical significance was considered when $p < 0.05$. Data are expressed as median (minimum—maximum).

**Table 7. Effect of dual ventilation with the ACRA interface of homogeneous (Step 1) and heterogeneous (Step 2) paired animal models on pH, partial pressure of arterial carbon dioxide, ratio of partial pressure of arterial oxygen to the fraction of inspired oxygen, percentage of hemoglobin saturated with oxygen, heart rate, mean arterial pressure, stroke volume, respiratory system compliance, positive end-expiratory pressure, driving pressure and expired tidal volume.**

| | STEP 1 | | | | STEP 2 | | | |
|---|---|---|---|---|---|---|---|---|
| | PIG 1 | | PIG 2 | | PIG 1 | | PIG 2 | |
| | No lung lavage | | No lung lavage | | No lung lavage | | Lung lavage | |
| | Mean (SD) | CI (95%) | Mean (SD) | CI (95%) | Mean (SD) | CI (95%) | Mean (SD) | CI (95%) |
| $PaO_2/FiO_2$ | 490.5 (12.0) | 382.5–598.5 | 517.0 (29.7) | 250.2–783.8 | 590.5 (31.8) | 304.6–876.4 | 147.5 (74.2) | -519.6–814.6 |
| $PaCO_2$ (mm Hg) | 59.0 (12.7) | -55.36–173.4 | 53.0 8.5) | -23.24–129.2 | 53.0 (7.1) | -10.53–116.5 | 67.0 (9.9) | -21.94–155.9 |
| pH | 7.315 (0.09) | 6.48–8.14 | 7.38 (0.05) | 6.94–7.83 | 7.34 (0.05) | 6.9–7.79 | 7.27 (0.04) | 6.88–7.65 |
| VTe (mL) | 183.5 (19.1) | 11.97–355 | 205.5 (21.9) | 8.55–402.4 | 189.0 (48.1) | -243–621 | 181.0 (27.9) | -60.42–422.4 |
| PEEP (cm $H_2O$) | 6.5 (12.1) | -12.56–25.56 | 3.5 (2.1) | -15.56–22.56 | 5.0 (0.0) | 5–5 | 12.5 (0.7) | 6.15–18.85 |
| DP | 14.5 (4.9) | -29.97–58.97 | 12.5 (2.1) | -6.56–31.56 | 12.5 (7.8) | -57.38–82.38 | 16.5 (0.7) | 10.15–22.85 |
| Crs (cm $H_2O$) | 12.5 (7.8) | -57.38–82.38 | 16.0 (0.0) | 16–16 | 13.0 (0.0) | 13–13 | 10.5 (0.7) | 4.15–16.85 |
| SpO2 (%) | 98.5 (2.1) | 79.4–117.6 | 99.5 (0.7) | 93.1–105.9 | 100.0 (0.0) | 100–100 | 99.5 (0.7) | 93.15–105.9 |
| HR (beats/min) | 71.0 (11.3) | -30.65–172.6 | 70.5 (19.1) | -101–242 | 72.5 (13.4) | -48.21–193.2 | 64.5 (16.3) | -81.62–210.6 |
| MAP (mm Hg) | 67.5 (10.6) | -27.8–162.8 | 88.0 19.8) | -89.89–265.9 | 89.5 (10.6) | -5.797–184.8 | 73.5 (12.2) | -34.50–181.5 |
| SV (mL) | 32.9 (3.4) | 1.13–64.67 | 42.5 (9.5) | -1.97–86.97 | 32.8 (1.1) | 22.64–42.96 | 37.8 (4.5) | -2.860–78.46 |

Data are expressed as mean (standard deviation) and are the average of the results of the two experiences. Confidence intervals (CI: 95%) are shown for each result.

ventilation, the use of a pinch valve imposes changes in the delivery of the volume and that tight and individual monitoring of ventilation (i.e., capnography) and precise setting of the ventilator alarms are paramount to ensure adequate volume delivery to each paired unit [11, 20, 21]. In our study, it was evident that the sole modification of the respiratory rate using a variable inspiratory time resulted in a decrease of the delivered volume to the paired unit in which a restriction to flow was applied. Although this result is considered as a drawback of dual ventilation [20], it could be offset by setting a fixed inspiratory time.

So far, the mainstay approach to overcome backflow during the respiratory cycle is the addition of one-way flow valves. In our experiment, the interposing of four one-way valves at key points of the novel interface was crucial to guarantee an adequate direction of flow both in inspiratory and expiratory phases and to differentiate Ppl and intrinsic PEEP of both paired units (Fig 1).

With the aim to control PEEP in each of the paired lung units in dual ventilation studies, the most commonly reproduced mechanism was to interpose an in-line and adjustable PEEP valve in each of the expiratory circuits, upstream to the one-way valves [8, 17]. The use of an adjustable in line PEEP valve, however, can result in inadequate pressure signals that lead to the ventilator not reaching its target PEEP, alarm triggering and undesirable ventilator responses. According to Roy et al. [11] and Raredon et al. [17], this could be offset by the addition of a 22 mm bypass circuit between the inspiratory and expiratory ports of the ventilator to ensure equal inspiratory and expiratory gas volume and pressure at the expiratory port of the ventilator. However, the addition of a bypass circuit could result in cross contamination [11]. To avoid this drawback, the interpose of a one-way valve in the bypass circuit is recommended [11]. In the ACRA interface, a 6 mm diameter, 15 cm length tube was used as bypass circuit. This was selected considering previously obtained results from our research group [22] and two important factors: 1) the flow resistance of the bypass circuit is proportional to the tube length and to the diameter elevated to the fourth power and, 2) the total air volume of the bypass circuit is proportional to the length and to the squared diameter. In our study, the selected tube permitted to operate with adequate pressure signals and at the same time

guaranteed a negligible back flow. Although the possibility of cross contamination was minimal, the likelihood of this tube being obstructed should not be discarded. Futures studies are warranted to further comprehend the relevance of the size of the bypass circuit.

In our study, we tested the placement of a single adjustable PEEP valve in only one of the paired units, the one that required the greater PEEP. This configuration permitted the ventilator to completely control the PEEP of the paired unit without an additional PEEP valve and set the minimal PEEP value of the unit with the additional adjustable PEEP valve. Different PEEP values for each paired unit can be achieved by adjusting the PEEP of the ventilator and the single adjustable PEEP valve. However, the interposition and non-simultaneous use of two adjustable PEEP valves could be desired when progression of the disease alters PEEP requirements. If the paired units progress to different PEEP requirements, the only possible solution using the ACRA is to exchange the expiratory limb and the adjustable PEEP to that who requires the highest PEEP. This is undoubtedly a weakness of the presented model that only highlights the difficulties inherent to dual ventilation. Further evaluations are warranted to determine if the alternant use of one of two interposed PEEP valves provides optimal management of individual PEEP in paired units.

Dual ventilation based on flow restriction introduces pressure drops along the respiratory system, which are independent and may not be in concordance with the ventilator setting. This discrepancy between the feedback pressure reading and the pressure set by the ventilator could either trigger an alarm and/or lead to the delivery of higher flow rates and excessively high tidal volumes to patients. In order to avoid this, the addition of a bypass circuit from the expiratory to the inspiratory limb bypassing the flow restrictor valves and patients, was necessary. With this scheme, the ventilator feedback control system detects a single 'virtual' patient with a small resistance (mainly the one of the bypass circuit) and a large compliance (mainly those of both patient lungs) and will adjust internal variables to obtain the predetermined setting. This feature maintains the ventilator setting, adds safety and has also been proposed by Raredon et al. [17].

In the face of the necessity of disconnection of one patient under dual ventilation, and in accordance with the suggested protocol for ventilator sharing [18], the ACRA permits a simple manual adjustment of the pinch valve that will guarantee zero flow to the circuit to be disconnected, maintaining current ventilation to the paired patient unaffected.

Individual monitoring with dedicated alarm setting should be considered as the optimal and safest option for patients under dual ventilation. Although challenging in a catastrophic situation in which shortage of medical equipment defines mortality rate and caregivers have to face a tough triage to ration medical goods [1, 2], the lack of individual respiratory mechanics monitoring should not preclude the application of dual ventilation as the last available resource. The interposing of the novel interface presented here resulted in the possibility to obtain the value of PIP, $P_{pl}$, PEEP and intrinsic PEEP of each paired patient through the lecture on the incorporated analogue manometers and to calculate their resulting ΔP. Although the ventilator screen displayed a value of $V_{Te}$ that corresponded to the summed volume of both units, $V_{Te}$ can be individualized. However, due to the several potential complications and those yet to elucidate, the use of individualized monitoring that includes capnography and VTe measurement should be a priority even under catastrophic scenarios [18, 21].

Our experiment was the first that used a preformed novel interface for dual ventilation that allowed individualized control of the driving pressure, in a live animal ARDS model. This study demonstrated that all the varied clinical scenarios tested first in lung units and then in breathing simulators, were successfully reproduced achieving lung protective ventilation strategy and acceptable gas exchange during dual ventilation. The testing of the ACRA on two paired animals with different respiratory mechanics contributed to the experiment with real-

time biological and hemodynamic responses from dual ventilation. Although the presence of asymmetrical $\Delta P$ and compliance between paired units could lead to adverse events such as auto-PEEP [22], the results obtained in this study could serve as a promising background in the face of shortage or for further investigations. It must be considered that although our animal model did not show significant differences on the compliance between paired units, the lung lavages were effective to simulate an ARDS scenario and the model considered useful to demonstrate that a different $\Delta P$ could be obtained with the use of the ACRA interface.

This experiment had several limitations. First, the ACRA did not allow us to independently control breathing rate and I:E ratio of the paired units and in our experiment all of the units were ventilated with the same $FiO_2$. Second, ventilator data as the sole source of $V_{Te}$ requires further analysis to obtain individual values. Third, expected variations in the morphology of the flow waveforms after flow was restricted were observed but not thoroughly analyzed. Fourth, paralyzing of the pigs was necessary to avoid spontaneous breaths that could have affected the paired unit. Fifth, our experiment was tested in a live animal model for only eight hours and limited scenarios were simulated in order to test the performance of the ACRA. Therefore, further experimental and clinical studies are guaranteed.

Undoubtedly, dual ventilation is a tool that should be used with precise criteria, appropriate training and individualized monitoring. It should be applied for the shortest time possible until a better option is available.

## Conclusions

In conclusion, it is with no doubt that dual ventilation limits the capacities of single ventilation. However, our experiment demonstrated that it is possible to independently control $\Delta P$ and $V_{Te}$ to each of the paired units. Additionally, the ACRA is a preformed unit that minimizes the risk of accidental misassemble adding a potential safety quality. In a pandemic situation in which shortage of ventilators can worsen health outcomes of the world population, this is an alternative worth to be considered.

## Supporting information

**S1 Video. The 3D printing models and additional information have been published on https://github.com/ACRA2020/Pinch_Valve_design.**
(MP4)

## Acknowledgments

The authors would like to thank Marcelo Campos, MD and Ariel Bonardi, BME from the 'Asociación de Anestesia, Analgesia y Reanimación de Buenos Aires' and Célica L. Irrazábal, MD from the 'Hospital de Clínicas José de San Martín' for their substantial collaboration with this project.

## Author Contributions

**Conceptualization:** Pablo E. Otero, Juan M. Cabaleiro, Guillermo Artana.

**Data curation:** Pablo E. Otero, Lisa Tarragona, Andrea S. Zaccagnini, Natali Verdier, Martin R. Ceballos, Emiliano Gogniat, Juan M. Cabaleiro, Juan D'Adamo, Thomas Duriez, Pedro Garcia Eijo, Guillermo Artana.

**Formal analysis:** Pablo E. Otero, Lisa Tarragona, Natali Verdier, Martin R. Ceballos, Emiliano Gogniat, Juan M. Cabaleiro, Juan D'Adamo, Thomas Duriez, Pedro Garcia Eijo, Guillermo Artana.

**Funding acquisition:** Pablo E. Otero, Guillermo Artana.

**Investigation:** Pablo E. Otero, Lisa Tarragona, Andrea S. Zaccagnini, Martin R. Ceballos, Emiliano Gogniat, Juan M. Cabaleiro, Juan D'Adamo, Thomas Duriez, Pedro Garcia Eijo, Guillermo Artana.

**Methodology:** Pablo E. Otero, Lisa Tarragona, Andrea S. Zaccagnini, Emiliano Gogniat, Juan M. Cabaleiro, Juan D'Adamo, Thomas Duriez, Pedro Garcia Eijo, Guillermo Artana.

**Project administration:** Pablo E. Otero, Guillermo Artana.

**Resources:** Pablo E. Otero, Guillermo Artana.

**Software:** Pablo E. Otero, Juan M. Cabaleiro, Juan D'Adamo, Thomas Duriez, Pedro Garcia Eijo, Guillermo Artana.

**Supervision:** Pablo E. Otero, Lisa Tarragona, Emiliano Gogniat, Juan M. Cabaleiro, Juan D'Adamo, Thomas Duriez, Pedro Garcia Eijo, Guillermo Artana.

**Validation:** Pablo E. Otero, Lisa Tarragona, Andrea S. Zaccagnini, Emiliano Gogniat, Guillermo Artana.

**Visualization:** Pablo E. Otero, Lisa Tarragona, Andrea S. Zaccagnini, Guillermo Artana.

**Writing – original draft:** Pablo E. Otero, Lisa Tarragona, Natali Verdier, Martin R. Ceballos, Emiliano Gogniat, Juan M. Cabaleiro, Juan D'Adamo, Thomas Duriez, Pedro Garcia Eijo, Guillermo Artana.

**Writing – review & editing:** Pablo E. Otero, Lisa Tarragona, Natali Verdier, Guillermo Artana.

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
