## [Decision Letter · Decision Letter 0]

2 Jun 2021

PONE-D-21-13259

Ventilator output splitting interface 'ACRA': description and evaluation in lung simulators and in an experimental ARDS animal model

PLOS ONE

Dear Dr. Otero,

Thank you for submitting your manuscript to PLOS ONE. After careful consideration, we feel that it has merit but does not fully meet PLOS ONE’s publication criteria as it currently stands. Therefore, we invite you to submit a revised version of the manuscript that addresses the points raised during the review process.

We look forward to receiving your revised manuscript.

Kind regards,

Aleksandar R. Zivkovic

Academic Editor

PLOS ONE

Journal Requirements:

Reviewers' comments:

Reviewer #1: Thank you for your work in this area. It is excellent to see an animal evaluation of these existing concepts.

Table 1 is difficult to understand and may be misleading. Please reconsider the way it is presented.

Can you please detail the exact parts used and provide access to the 3D models for the enclosure? There is currently insufficient information given to reproduce the experiment accurately.

FDA approval is a US-centric concept, however this study was performed in Argentina. Are the devices used in this apparatus available elsewhere?

It appears that there was an attempt to upload the data to OSF.io, but the files are all currently zero bytes in length: has an error been made in the upload process? Is this an error with OSF.io? Regardless, the design files for the box do not appear to be available, nor is the bill of materials.

The schematic layout of the ACRA design does not seem to match the renders. In the renders, the PEEP valve appears to be on the inspiratory ports. Are the labels on the diagrams correct?

Can you please elaborate on the engineering basis for the "short-circuit tubing"? This does not appear to be required for all ventilators. Can you please specify how the diameter of 6mm was selected?

Figure 3: Confidence intervals would be preferable and would emphasis the uncertainly inherent in the extremely small sample size. It may be clearer to present this data in a table and to specify which of the pigs was the one on whom the lavage intervention was performed.

Conclusion: The first sentence "In conclusion, it is with no doubt that dual ventilation limits the capacities of single ventilation." is unclear. Perhaps you mean "Shared ventilation for two patients with one ventilator is possible, but there is limited ability to modify the ventilation parameters. Our experiment demonstrated one approach to modify the driving pressure and PEEP delivered to each patient in an animal model."

Reviewer #2: Overall: The authors present benchtop data to support a circuit design that is able to ventilate two “patients” with divergent lung compliances/PEEP needs with one ventilator, and preliminary testing in an animal model of ARDS. This is presented in three phases. The phase 1 data are qualitative and not scientifically sound. They are worthy of presentation and discussion, but unless there are added details about how these were recorded and/or quantitative measurements available, this phase should be addressed only as proof of concept testing that aided in initial design of the circuit, not as data supporting the circuit’s adequate performance. Phase 2 is the most robust and does provide evidence that pinch restriction allows differential ventilation. Phase 3 tests the circuit in 2 sets of pigs with different degrees of ARDS. Unfortunately, 2 experiments does not allow sound scientific comparison, and the ARDS induced does not appear to result in widely different lung compliances, and so does not allow full demonstration of the capability of the circuit to match tidal volume with different ventilatory needs (the PEEP/oxygenation data, however, are encouraging). Finally, as tested here, the short circuit feature, which lacks a one-way valve to prevent cross-contamination between patients, is a safety concern (in addition to the overarching safety concern of shared ventilation, of course). This must be corrected before this circuit is presented to the scientific/clinical community.

The primary question that comes to mind, apart from the scientific issues above, is why are two flow restrictor valves needed? Would it not be simpler (and more resource sensitive, in resource limited settings) to apply and adjust just one valve for the “lower compliance” patient, and overall ventilator settings for the “high compliance” patient? In this way, at least one of the inspiratory pressures read on the ventilator would (relatively) correctly reflect the pressures delivered to the patient. This may also remove the need for the short circuit to keep the ventilator functioning.

In terms of readability/intelligibility, the sentence structure and grammar is of good quality (although not perfect; this reviewer commends authors writing in a second language, and suggests simply that a native English speaker could correct minor issues with this). The introduction and discussion are thorough (but not overly detailed) and flow well, However, the organization of the methods and results is poor and needs improvement – comments below.

Abstract:

What does the keyword “interphase” refer to? Is “interface” what is meant?

Methods:

Information on the components/manufacturers for circuit parts is needed if this is to contribute to scientific knowledge and/or aid clinician in a time of crisis. P20 L310-313 suggests that this unit is preformed and easily reproducible, but that is only the case if components are better specified.

Why analog manometers, when digital manometers can just as easily quantify and trend PIP, and be displayed on a monitor (without the need to be at patient bedside in the case of infectious disease or other risk to personnel)? Are these analog manometers more available in low resource settings? Discussion and instructions for assembly should include an option for digital manometers, which may actually be more readily available in many healthcare settings, where they are frequently used in critically ill patients for invasive pressure monitoring. This has been suggested previously: Cherry, Anne D et al. “Shared Ventilation: Toward Safer Ventilator Splitting in Resource Emergencies.” Anesthesiology vol. 133,3 (2020): 681-683. doi:10.1097/ALN.0000000000003410

The description of the short-circuit tubing P10 L85-87 is unclear at the time of first introduction. This is left open when both circuits are operational, it sounds like? The reason for it is also not discussed until the discussion, but may not be intuitive for readers. The short circuit proposed in reference (Raerdon et al. “ Pressure-Regulated Ventilator Splitting (PReVentS) – A COVID-19 Response Paradigm from Yale University” https://doi.org/10.1101/2020.04.03.20052217) has a one-way check valve, but that is not included here and may be important (below).

The simulation of disconnects and inspiratory efforts is a nice addition. How is it possible that disconnection of sim 2 (or sim 1) does not affect ventilation of the pair (P18 L253)? Is this disconnection with capping of the ventilator tubing? Without a cap (or the “full pinching” suggested later, in the discussion), it is difficult to imagine that disconnection would not result in total loss of ventilation to both units. What is the effect of/procedure for when there is a suctioning procedure (temporary increased resistance in tubing), downstream occlusion of any portion of the ventilator tubing distal to the endotracheal tube, or patient/ventilator dys-synchrony other than inspiratory effort (coughing, valsalva)? It appears that expiratory limb occlusion near the ventilator would actually result in expiratory gas flow across the short circuit and back to the inspiratory limb to the patients(s)? Similarly, if one patient coughs during inspiration? This would result in a cross-contamination between patients. Were these scenarios simulated with this circuit setup?

P11 L116 what is “each unit's respiratory mechanics monitor”? With what method(s) was flow/tidal volume measured (other than phase II, since the ASL 5000 has integrated data output)? The ACCU Lung does not appear to have integrated data outputs, but the methods, as written, imply that data were recorded (see also similar comments about table 1). P13 L 147 states that VTE changes for phase II were measured by occluding the ETT and compared to ASL 5000 data outputs…was this method ETT occlusion method used for phases I also (why not, if not)?

The methods and results are somewhat mixed, with table 2 presenting results of phase II within the methods section. Table 3, meanwhile, presents a comparison of measurement methods for phase II. These should be reversed. Table 1 is discussed in the results, yet it appears in the methods. Also in terms of organization and readability, elements that are important for understanding the methods (such as details about test lung manufacturers, maneuvers for testing the lung units) are scattered throughout the text and make it difficult to determine precisely what was done and how it was done in each testing phase.

Table 2 is confusing. The majority of the text discusses split ventilation, but at first glance and after reading the table legend, this table appears to test each lung simulator separately, with the equivalent of some sort of pressure limited, volume control mode? Since it is presented under methods the reader immediately assumes it is somehow presenting characteristics of each test lung individually. Otherwise it is not clear how Sim1 “sees” a PIP pressure of 15 while the overall ventilator is set to a PIP of 35 or 40, unless these are actually “results” of using the resistance valve to adjust airflow (this is the case, presumably).

Results:

The data files uploaded into the OS database are, without exception, corrupt, unreadable, or appear to be empty CSV files when this reviewer tried to access them. Revision/reuploading is needed.

Is table 1 presumed changes or were there actual measurements made? The table caption says “proposed” but the text P 12 L17 says “recorded”. If they were recorded, were they quantified and/or were the “recorders” blinded somehow? PEEP seems particularly difficult to estimate without a quantitative measurement, yet P17 L 238 states that PEEP could be set without impacting PEEP on the other lung unit – how were changes in PEEP determined? If the “eyeball” test was used throughout phase I, this should be stated. As written, it sounds as though they were measured, but the methods of measurement are not reported and qualitative results are presented.

Was the short circuit tubing occluded for the 5 second VTe determinations? Does the short circuit tubing impact the accuracy of this method of measurement as a surrogate for what’s happening when the entire dual ventilation unit is in line on both lung units?

P19 L 270-272 “the required adjustment on the ventilator setting (e.g. increased f) imposed changes (e.g. reduced VTe to maintain minute ventilation) on the other model.” Is unclear. Is f frequency (i.e. respiratory rate)? That is the first time this is mentioned in this document.

The animal model is an excellent addition, but with only two sets of animals, conclusions are extremely difficult to draw and statistical comparison is not presented (there appears to be a considerable amount of heterogeneity, so perhaps no comparisons demonstrated significant differences). There does appear to be a profound difference in the two “degrees” of ARDS in terms of P/F ratio, but no clear/consistent difference between them in the important variable most discussed in this manuscript, compliance. As such, the differential in applied PEEP is of interest, but the differences (or lack thereof) in other respiratory parameters (achieving an equivalent VTe, for example) is less exciting. Perhaps the manuscript could focus somewhat more on the manipulation of PEEP as the novel finding. Understandably, in the situation where ventilator splitting is needed, titration of both settings is necessary so both elements do need to be discussed.

Animals of different size + degree of ARDS may have given a more robust difference between compliance/tidal volume needs, and would have more dramatically illustrated the capability of this circuit to provide differential ventilation and maintain the ventilator parameters presented.

Conclusions:

The effect of the adjustable resistance on tidal volume will vary depending on respiratory rate/insp time; these effects must be discussed, stressing that resistance would need to be adjusted if ventilator respiratory rate or inspiratory time settings were changed, even if inspiratory pressure settings remained the same (also discussed in Cherry, et al). If the investigators could provide test data for varying respiratory rates with their setup, it would add significant novelty and interest.

Reviewer #3: The authors have developed an interesting device and are commended on their work. The article is well written and does address some important aspects of differential multiventilation. However, there are some very important problems with the manuscript that must be addressed. The most important issues are that some of the conclusions are directly contradicted by works the authors reference as well as other available literature, the study has some critical limitations that limit some of their conclusions, and that the authors present their findings primarily in a qualitative manner and do not present quantitative results. Finally, while the authors link to their data, the files do not contain any data that could be used to validate their findings. Again, the authors have conducted some relevant and important work that would likely be of interest to the global community once these largely resolveable problems with the manuscript are addressed.

Major conceptual issues:

The authors link to their data, however the files do not contain any data that could be used to validate their findings and the OSF websites lists them as "Table empty or corrupt." The data should be reuploaded ensuring the character encoding of their comma-delimited files are compatible with the OSF site.

It is not clear that the shutoff valve in the short-circuit tubing is necessary and it's presence raises several concern. If the valve were accidentally closed, it would dramatically change the behavior of the device due to the absence of a short-circuit or bypass-circuit. The full implications are reviewed in by Roy et al. last year (Crit Care Explor, 2020, 10.1097/CCE.0000000000000198), but in brief, these could include:

-potentially ramping up the inspiratory pressure in both circuits,

-causing unpredictable behavior,

-triggering the obtruction alarm,

-or worst, causing no detectable problem immediately, caregivers leave the room with both patients alone in an isolation room with a device configuration that will suddenly causes the ventilator to behave erratically only once another problem occurs (e.g. single patient circuit obstruction).

Another problem with the short-circuit is that, because there is no one-way valve in the short circuit, expired CO2 or pathogens may potentially recircuilated through the circuit (particularly if respiratory effort is made) and be rebreathed by one or both patients. The authors did not test for this potentially crucial pitfall and very concerning possibility. Therefore, if the valve is left open during regular use, then this major potential issue must be discussed in the paper. If the valve is closed during regular use, then the previous issue becomes an even more major concern.

On line 330, the authors state that the "simultaneous use of two PEEP valves, however, can result in inadequate pressure signals that lead to the ventilator not reaching its target PEEP, alarm triggering and undesirable ventilator responses and would be, therefore, not recommended." There are two problems with this statement: First, it is not technically correct. The predictable behavior of the ventilator depends on the ventilators ability to self sense. The problems described by the authors are seen when there is no adequate bypass circuit AND there is modification of either inspiratory or expiratory pressures in BOTH circuits. If the valve on their device's short-circuit is closed and/or the tubing provides inadequate flow (i.e. due to the small tubing with a narrow opening through the valve), the authors will likely still see such behavior when the PEEP valve is set to higher settings and the pinch valve of the other circuit is set sufficiently closed. These implications are described in detail in the Roy et al. paper (Crit Care Explor, 2020, 10.1097/CCE.0000000000000198). Moreover, the second problem with the statement on line 330 is that the authors have, in their manuscript, cited Raredon et al. who provided a very thorough testing of device with exactly the design the authors claim to be problematic. Not only did Raredon et al not see this behavior, but they directly contradict the authors statement and explictly describe the opposite. At very least, the authors must address in the discussion section how such a major descrepancy could exist between their findings and the contradictory findings of Raredon et al.

Table 1 provides a qualitative overview of what should happen with use of the author's device. However, there are two problems in the manuscript that are particularly noticed in this table: first, the table is presented as a table of results, in which case quantitive rather than qualitative results are appropriate. As in other parts of the paper, the authors report qualitatively summarized results when detailed quantitative data would be more appropriate. The second, perhaps larger issue is the methodological: the loosely described protocol appears to involve manipulation of only one variable at a time. Manipulation of only one variable at a time does not allow for the interaction between components. As outlined above, there are major possible (even likely!) interactions that could occur here and they have been described in detail in the literature. In fact, such interactions would have been very likely occur in the author's device during testing since the short circuit was occluded (and perhaps, even if it had not). The authors must provide supporting data for their conclusion, address how their protocol (including all of the tests) would detect interactions between components, and discuss any remaining limitations in the protocol.

Table 2 demonstrates that only a fixed PIP, a fixed PEEP, a fixed respiratory rate and a fixed I:E ratio were used for testing. Because a pinch valve is used in the inspiratory circuit rather than a pressure release valve (as others have used), the pressure drop across the pinch valve will change based on the Inspiratory time, the ventilator set PIP & PEEP, and the setting on the PEEP valve. This major caveat results in non-linear changes in the PIP when any of these 3 settings are changed. By using only a single fixed ventilator setting for the testing, none of the effects that result from this caveat are able to be observed. In real use, it is likely that both pinch valves may need adjusted anytime a change is made to the I:E ratio, the I time, the PIP, or the PEEP. These important non-linear effects of ventilator setting changes is not seen in the authors' testing, but remains highly relevant to the use of the device. Ideally, the testing protocol would be more developed and address these issues. At minimum, these caveats and their potential repercussions must be discussed in the appropriate section.

Other issues

abstract: The abstract does not include any of the results. Similar to the body of the manuscript, the authors articulate success (e.g. "feasible") rather than results ("consistent","stable","unaltered", etc). Please add a sentence that summarizes the measurement results i.e. what outcome measure(s) indicated that it was feasible.

line 23: The authors states the components are all "approved", but it is not clear who has approved them and is doubtful that all were approved for use in this manner. Approval is given to medical devices for specific uses. For example, many countries require separate certification for materials that are in the breathing pathway (e.g. ISO 18562). Are the parts all approved for use within the gas path according to such a specification? Who approved them for such use?

line 25/figure 2: The PEEP valve is place on the expiratory limb of the unit that would require the greater PEEP. However, what happens if the individual who requires the greater PEEP changes? The authors mention in the discussion that this might occur, but do not address how it would be fixed... Is it possible to switch over the valve or does this require disassembly?

line 330: the authors cite 3 papers to support the notion that inline PEEP valve may be used to modify the PEEP. The cited article by Clarke et al. does not support this statement as it does not discuss inline PEEP valves at all. Additionally, the author's device appears to utilize a particular type of inline modification of a commercial PEEP valve described by Bunting et al. (AJEM, 2020, 10.1016/j.ajem.2020.06.089) which was not cited.

line 370 & 390: V Te can be individualized but only indirectly through changes in pressure. It should be stated clearly that changes in Vte are can only be individualized by means of pressure adjustment and use of such a device in a Volume Control mode would be dangerous.

Reviewer #4: This paper by Otero et al. discusses an interesting new device to enable a more accurate control of shared ventilation. Shared ventilation is mostly discouraged by most professional societies (as also discussed by the authors) permitting it only in specific cases and while ventilator shortage does not seem to be as a major issue as it was at the beginning of the pandemic, the concept of ventilation sharing can still be applied in emergency settings and extra means to promote the safety of this practice is welcome in my opinion.

My questions and opinions are the following:

1) Did the authors record or assess the flow patterns of the individual circuits?

2) This device seems to manipulate the resistance of the individual circuits by the pinch valves to compensate for the differences in compliance, resulting in impedances with a suitable ratio to deliver appropriate tidal volumes. In this regard, the principle behind this device seems to be somewhat similar to another study recently published (doi: 10.1016/j.resp.2020.103611). This should be discussed in the manuscript.

3) The use of manual pinch valves can be problematic if patient dynamics change, since it seems that the system cannot deliver alerts in case ventilation to the specific patients change. This can mean an even higher burden on the ICU staff already under high pressure to actively monitor the manometers, in my opinion actually restricting the use of the device more than the need for ventilators in most ICUs. The single extra PEEP valve can also pose some risks: as the authors describe it should be place in the circuit which seems to require a higher PEEP. However, the course of the disease can be quite different even in patients that are similar at the time of recruitment, resulting in a change of “higher PEEP circuit”, requiring the complete swap of the circuits and re-adjusting all the pinch valves to deliver appropriate volumes.

4) In Table 1 the authors describe that increasing PEEP in Lung 2 results in an increased PEEP in Lung 1 and no change in Lung 2. I presume this is a typo and the authors meant to describe the effects of a PEEP increase on Lung 1. While having constant settings on the ventilator, setting the pinch valve on one lung should also have an effect on the ventilation of the other circuit, since the ratio of impedances between the two circuits is going to be altered, resulting in a different split of delivered volume. However, the authors describe no effects on the other circuit (Table 1, Pinch valve on Lung 1 and 2).

5) I don’t really see how the occlusion of one circuit to measure the volume won’t affect the delivered volume to the other end, since the practically infinitely high resistance of the blocked ET tube should reroute some of the volume to the open lung, since the total compliance of the circuit should be different. Did the authors directly measure each tidal volume by placing flow meters into the individual circuits to assess this or did they assess the individual tidal volumes only using this method? I have also similar doubts about the lack of effect on the other patient in case of a disconnection, since the open tube of the disconnected patient should also change the flow distribution. If the valve of the patients is closed (in case of a planned disconnection), then the ventilator should be somewhat adjusted for the changed total compliance of the circuit.

6) While the use of the short circuit tubing is a viable method to trick the ventilator to using different amounts of PEEP, it also carries some risks due to the lack of some alerts and also since this short circuit is actually a low-impedance third patient that would also need to be balanced according to the actual clinical scenarios in the two real patients. This would necessitate a third flow meter to verify that the short circuit tubing is not stealing considerable amounts of air from the real patients.

7) There seems to be a typo on Fig. 2: the captions states that #6 is the PEEP valve and it seems to be the case based on the photo as well. However, while the caption states that #7 and #7’ are the connections for the expiratory tubing and #8 and #8’ are the connections for the inspiratory tubing, based on the pictures this should be the opposite in my opinion, with #7 to be the inspiratory and #8 to be the expiratory port.

8) What kind of ICU ventilators did the authors use? Did they face any alerts from the ventilator due to the use of the splitter device?

6. PLOS authors have the option to publish the peer review history of their article (what does this mean?). If published, this will include your full peer review and any attached files.

Reviewer #1: **Yes: **Alexander Linden Clarke

Reviewer #2: No

Reviewer #3: No

Reviewer #4: No

---

## [Author Response · Author response to Decision Letter 0]

5 Aug 2021

Dear reviewers,

Thank you very much for your time, comments, and valuable suggestions. The reviews have challenged our study, and the corrections have resulted in a better product. We hope that this version will meet the publication criteria and that we have clarified the points raised by the reviewers in the following answers.

We want to apologize for the delay in responding and organizing the new version of the manuscript. Quarantines, complications in times of pandemics have conspired against the organization of the team. Apologies for that!

All the best, 

The authors

Reviewer #1: 

Thank you for your work in this area. It is excellent to see an animal evaluation of these existing concepts.

Response: Thank you very much for the time taken to review our manuscript and our sincere apologies for the time taken to respond to the revision. 

1. Table 1 is difficult to understand and may be misleading. Please reconsider the way it is presented.

Response: Table 1 was modified by the addition of the qualitative data obtained. Other reviewers had also proposed modifications to the table, and we hope that this is clearer and explanatory to the reader. 

2. Can you please detail the exact parts used and provide access to the 3D models for the enclosure? There is currently insufficient information given to reproduce the experiment accurately.

Response: Yes, absolutely. The 3D models (.STL files) have been added to the repository. Link: https://github.com/ACRA2020/Pinch_Valve_design

The supplementary information (repository) is available on-line in a Dropbox link: https://www.dropbox.com/sh/nys8q88rupr3b3o/AAAog1Rq9iGrv97xW3GkAQc4a?dl=0

3. FDA approval is a US-centric concept, however this study was performed in Argentina. Are the devices used in this apparatus available elsewhere?

Response: This is true. Indeed, in Argentina the national and official regulatory agency is called ANMAT and works in network with the FDA, as well as the agencies of other countries. Therefore, FDA regulations indirectly apply to the regulations of other countries, and this was our case. A protocol of use has been developed and approved by the ANMAT, given that the parts that constitute it were previously FDA approved. We assume that the components used are widely available in the world. 

4. It appears that there was an attempt to upload the data to OSF.io, but the files are all currently zero bytes in length: has an error been made in the upload process? Is this an error with OSF.io? Regardless, the design files for the box do not appear to be available, nor is the bill of materials.

Response: We are not sure what happened during the upload process but we are sure that the files are correctly uploaded and will be available. Our apologies for this technical problem.

5. The schematic layout of the ACRA design does not seem to match the renders. In the renders, the PEEP valve appears to be on the inspiratory ports. Are the labels on the diagrams correct?

Response: Thank you very much for this observation. This was a mistake on the numbers on the picture which we have already amended. 

6. Can you please elaborate on the engineering basis for the "short-circuit tubing"? This does not appear to be required for all ventilators. Can you please specify how the diameter of 6mm was selected?

Response: A short circuit tubing between inspiratory and expiratory limbs ensures compatibility of the system with different ventilator controllers. This was also proposed by Raredon MSBet al: Pressure-Regulated ventilator splitting (PReVentS): A COVID-19 response paradigm from Yale University. medRxiv 2020; doi: 2020.04.03.20052217. And in the paper of Roy et al. an elegant discussion of this issue is also present. We added this reference too. 

As the ACRA system introduces pressure drops in the respiratory circuit that are unknown to the ventilator controller, the feedback loop that enforces the different aim pressures throughout a respiration cycle might spiral out of control if the system never reaches its target. This is the case while measuring the target PIP from the expiratory circuit, as the pressure drop introduced by ACRA imposes an effective PIP which might be below target by a noticeable margin. Depending on the monitoring enforced this could either set off an alarm, or simply lead to a feedback loop in the inlet regulation that provides a higher flow rate and, ultimately, that delivers excessively high volumes to patients. Additionally, a great discrepancy between inspiratory and expiratory pressures could lead to monitoring errors or false alarms (Raredon et al 2020). Therefore, to establish an adequate protocol that is independent of both the ventilator used and the feedback pressure probe location, it is beneficial to use a bypass circuit (e.g., our short-circuit) inspiratory and expiratory circuits as indicated in Figure 1 (green line in single-line diagram) with a small size tubing (i.e. 6 mm in diameter). 

With this scheme, the ventilator feedback control system senses a single `virtual' patient with a small resistance (mainly that of the short-circuit tubing) and a large compliance (mainly those of both patient lungs). Hence, the controller will target the parameters fixed of the monitor and adjust internal variables to obtain the predetermined goal of inspiration as a consequence of any disturbance with this patient configuration.

The need of this tubing was observed during the experiments using the Nellcor 65 Puritan Bennet 760 ventilator. However, additional tests of this device using other ventilators (data not presented in manuscript but available as data in the repository) did not show this requirement. It is not possible to test this device with all ventilators available and for this reason, initially, we provide the clinician with the option to close the short-circuit tubing or leave it opened, depending upon necessity. 

We looked for a hydraulic design of the tube that reduced the possibility of cross contamination between patients (during expiration) and at the same enabled the possibility to almost equalize pressures between limbs during inspiration.

The hydraulic predesign of the size of this tube was performed considering two important factors: 1) The flow resistance of the by-pass circuit is proportional to the tube length and to the diameter elevated to the fourth power and, 2) the total air volume of the short-circuit tube is proportional to the length and to the Diameter^2. 

The verification of the adequate size of this tubing (length and diameter) was undertaken using the numerical model described in (Eijo PMG, D’Adamo J, Bianchetti A, et al (2021) Exhalatory dynamic interactions between patients connected to a shared ventilation device. PLoS ONE 16, e0250672. doi: 10.1371/journal.pone.0250672). A typical result obtained with this model is shown in the figures below.

As we can see, the use of this model showed us that there exists an almost negligible flow from expiration to inspiration limbs. The integration of this flow along time results in the total air volume associated with this flow (2.9 mL), which is lower than the tube volume (5.6 mL) (red area in the right figure below). This small fraction of the volume of the tube is washed during the inspiration and replaced with fresh air coming from the ventilator.

From this analysis we concluded that the choice of the size of the tubing was correct for the purpose to avoid cross contamination.

Figure: Left) Ventilator pressure vs time, Middle) Pressure Difference between the ends of the tubing during time, Right) Flow rate through the tube against time (positive values from inspiration limb to expiration limb) 

7. Figure 3: Confidence intervals would be preferable and would emphasis the uncertainly inherent in the extremely small sample size. It may be clearer to present this data in a table and to specify which of the pigs was the one on whom the lavage intervention was performed.

Response: Thank you for the suggestion. The figure was now converted into a Table, and confidence intervals were added, as well as indication on which animal was performed the lung lavage. 

8. Conclusion: The first sentence "In conclusion, it is with no doubt that dual ventilation limits the capacities of single ventilation." is unclear. Perhaps you mean "Shared ventilation for two patients with one ventilator is possible, but there is limited ability to modify the ventilation parameters. Our experiment demonstrated one approach to modify the driving pressure and PEEP delivered to each patient in an animal model."

Response: Thank you very much for the suggestion. This sentence was amended as requested and we hope it is clearer.

Reviewer #2: 

Overall: The authors present benchtop data to support a circuit design that is able to ventilate two “patients” with divergent lung compliances/PEEP needs with one ventilator, and preliminary testing in an animal model of ARDS. This is presented in three phases. 

The phase 1 data are qualitative and not scientifically sound. They are worthy of presentation and discussion, but unless there are added details about how these were recorded and/or quantitative measurements available, this phase should be addressed only as proof of concept testing that aided in initial design of the circuit, not as data supporting the circuit’s adequate performance. 

Thank you for your work in revising our manuscript. We have amended the way data of Phase 1 is presented (now is quantitative), and we hope this fulfils your requirements. 

Phase 2 is the most robust and does provide evidence that pinch restriction allows differential ventilation. 

Thank you very much. 

Phase 3 tests the circuit in 2 sets of pigs with different degrees of ARDS. Unfortunately, 2 experiments does not allow sound scientific comparison, and the ARDS induced does not appear to result in widely different lung compliances, and so does not allow full demonstration of the capability of the circuit to match tidal volume with different ventilatory needs (the PEEP/oxygenation data, however, are encouraging). 

We absolutely agree with you. The model presented here did not manage to prove capabilities over a wide range of compliances and although we obtained a reduction of compliance of over 30% in both paired distressed animals, this reduction and the small sample size was not large enough to obtain an ideal model. Of note is, however, that the delivered PIP and PEEP were different between paired animals, which is actually the main objective of this study. We believe that this study could serve at least as a basis for future research and discussions. We hope you can support us in this. 

Finally, as tested here, the short circuit feature, which lacks a one-way valve to prevent cross-contamination between patients, is a safety concern (in addition to the overarching safety concern of shared ventilation, of course). This must be corrected before this circuit is presented to the scientific/clinical community.

Initially we have had this same concern, but further (unpublished) experiments by our research group demonstrated that the flow in this 6 mm short-circuit tubing during expiration is minimal, and therefore the risk of cross contamination is null. Nevertheless, filters downstream the shared device were added to minimize this risk. The addition of one-way valves has been proposed when the short-circuit tubing was of larger bore. We will however reconsider this recommendation and present the new device with the addition of a one-way valve. 

1. The primary question that comes to mind, apart from the scientific issues above, is why are two flow restrictor valves needed? Would it not be simpler (and more resource sensitive, in resource limited settings) to apply and adjust just one valve for the “lower compliance” patient, and overall ventilator settings for the “high compliance” patient? In this way, at least one of the inspiratory pressures read on the ventilator would (relatively) correctly reflect the pressures delivered to the patient. This may also remove the need for the short circuit to keep the ventilator functioning.

Response: This sounds absolutely logic. We proposed the use of two flow restrictor valves to independently manage the delivered pressures (PIP) to each of the paired patients. If there was only one flow-restrictor valve, every desired adjustment on the patient without this valve would require modifications on the ventilator that would inevitably lead to further adjustments on the flow restrictor valve of the paired unit. In conclusion we believe that the presence of two flow-restrictor valves would allow fine, precise and relatively individualized control of the ventilation of each unit. 

The presence of the short-circuit tubing is not linked to the use of two flow-restrictor valves, but to the ventilator controller that could target to equalize pressure between limbs during inspiration. 

1. In terms of readability/intelligibility, the sentence structure and grammar is of good quality (although not perfect; this reviewer commends authors writing in a second language, and suggests simply that a native English speaker could correct minor issues with this). The introduction and discussion are thorough (but not overly detailed) and flow well, However, the organization of the methods and results is poor and needs improvement – comments below.

Response: Thanks for your comments. We hope the new version is clearer

1. Abstract:

What does the keyword “interphase” refer to? Is “interface” what is meant?

Response: Yes, thank you for pointing out this typo. This was amended. 

1. Methods:

Information on the components/manufacturers for circuit parts is needed if this is to contribute to scientific knowledge and/or aid clinician in a time of crisis. P20 L310-313 suggests that this unit is preformed and easily reproducible, but that is only the case if components are better specified.

Response: Yes, absolutely. The 3D models (.STL files) have been added to the repository. Link: https://github.com/ACRA2020/Pinch_Valve_design

The supplementary information (repository) is available on-line in a Dropbox link: https://www.dropbox.com/sh/nys8q88rupr3b3o/AAAog1Rq9iGrv97xW3GkAQc4a?dl=0

1. Why analog manometers, when digital manometers can just as easily quantify and trend PIP, and be displayed on a monitor (without the need to be at patient bedside in the case of infectious disease or other risk to personnel)? Are these analog manometers more available in low resource settings? Discussion and instructions for assembly should include an option for digital manometers, which may actually be more readily available in many healthcare settings, where they are frequently used in critically ill patients for invasive pressure monitoring. This has been suggested previously: Cherry, Anne D et al. “Shared Ventilation: Toward Safer Ventilator Splitting in Resource Emergencies.” Anesthesiology vol. 133,3 (2020): 681-683. doi:10.1097/ALN.0000000000003410

Response: The main reason for choosing analog manometers was availability in our setting and low cost. Additionally, an analog manometer has the advantage that it is simple, robust and does not require power source. In our study, we have demonstrated that the reading between analog and digital devices was comparable. We thank you for the comment and for the shared reference. We will consider this suggestion for future developments of the device. 

1. The description of the short-circuit tubing P10 L85-87 is unclear at the time of first introduction. This is left open when both circuits are operational, it sounds like? The reason for it is also not discussed until the discussion, but may not be intuitive for readers. The short circuit proposed in reference (Raerdon et al. “ Pressure-Regulated Ventilator Splitting (PReVentS) – A COVID-19 Response Paradigm from Yale University” https://doi.org/10.1101/2020.04.03.20052217) has a one-way check valve, but that is not included here and may be important (below).

Response: Thanks for your suggestion. The text has been modified. We hope it is more explanatory.

NOTE: The use of a one-way valve would be helpful if the connecting tube was oversized, as suggested by Raredon et al. However, in our model the tube has an internal diameter (e.g. 6 mm) that prevents kinked flow. 

We modify the text in order to be more explanatory. 

1. The simulation of disconnects and inspiratory efforts is a nice addition. How is it possible that disconnection of sim 2 (or sim 1) does not affect ventilation of the pair (P18 L253)? Is this disconnection with capping of the ventilator tubing? Without a cap (or the “full pinching” suggested later, in the discussion), it is difficult to imagine that disconnection would not result in total loss of ventilation to both units. What is the effect of/procedure for when there is a suctioning procedure (temporary increased resistance in tubing), downstream occlusion of any portion of the ventilator tubing distal to the endotracheal tube, or patient/ventilator dys-synchrony other than inspiratory effort (coughing, valsalva)? 

Response: Regarding the case of a disconnection, we agree that this concept may elicit doubts. We have noted that not all ventilators can compensate for this loss. In our experiments, this was compensated and corroborated (additional data was uploaded). 

Please check the video we have provided.

1. It appears that expiratory limb occlusion near the ventilator would actually result in expiratory gas flow across the short circuit and back to the inspiratory limb to the patients(s)? 

Response: Although this hypothetical and unlikely situation could occur, we consider the addition of a one-way valve in the short circuit tubing in future developments. The manuscript was amended in order to make clearer the presence and functioning of the short-circuit tubing. 

2. Similarly, if one patient coughs during inspiration? This would result in a cross-contamination between patients. Were these scenarios simulated with this circuit setup?

Response: Further explanations on how this is mitigated by the presence and functioning of the short-circuit tubing was added to the text. Cross-contamination through this small-bore tube is unlikely, and is further minimized by the presence of filters. A coughing scenario was considered but not simulated. 

NOTE: We are aware that most literature on this topic uses a 22 mm by-pass circuit. However, we chose to use a 6 mm diameter and 15 cm length tube. This was selected taking into account previously obtained results from the numerical model described in (Eijo PMG, D’Adamo J, Bianchetti A, et al (2021) Exhalatory dynamic interactions between patients connected to a shared ventilation device. PLoS ONE 16, e0250672. doi: 10.1371/journal.pone.0250672), from our same research group. In order to select the size of the tube, two important factors must be taken into account: 1) The flow resistance of the by-pass circuit is proportional to the tube length and to the diameter elevated to the fourth power and, 2) the total air volume of the short-circuit tube is proportional to the length and to the Diameter^2. In our study, the selected tubing guaranteed a negligible flow from expiration to inspiration limbs. The integration of this flow along time results in the total air volume associated with this flow (2.9 mL), which is lower than the tube volume (5.6 mL). In our numerical tests, the possibility of cross contamination between patients associated with this flow was almost null. Please refer to the figure below. 

Figure: Left) Ventilator pressure vs time, Middle) Pressure Difference between the ends of the tubing during time, Right) Flow rate through the tube against time (positive values from inspiration limb to expiration limb) 

We understand that the use of a one-way valve would have been beneficial, and therefore we will take this into consideration for future developments of the ACRA device. This was also added to the text. 

1. P11 L116 what is “each unit's respiratory mechanics monitor”? With what method(s) was flow/tidal volume measured (other than phase II, since the ASL 5000 has integrated data output)? The ACCU Lung does not appear to have integrated data outputs, but the methods, as written, imply that data were recorded (see also similar comments about table 1). P13 L 147 states that VTE changes for phase II were measured by occluding the ETT and compared to ASL 5000 data outputs…was this method ETT occlusion method used for phases I also (why not, if not)?

Response: Thank you for highlighting this point. The text was modified. In fact, a monitor was used for each lung unit to individualize the variables. We apologize for the omitted details of the monitor used in the original manuscript.

1. The methods and results are somewhat mixed, with table 2 presenting results of phase II within the methods section. Table 3, meanwhile, presents a comparison of measurement methods for phase II. These should be reversed. Table 1 is discussed in the results, yet it appears in the methods. Also in terms of organization and readability, elements that are important for understanding the methods (such as details about test lung manufacturers, maneuvers for testing the lung units) are scattered throughout the text and make it difficult to determine precisely what was done and how it was done in each testing phase.

Response: Thanks for your suggestions. The text was amended. We hope it is more verbose now.

1. Table 2 is confusing. The majority of the text discusses split ventilation, but at first glance and after reading the table legend, this table appears to test each lung simulator separately, with the equivalent of some sort of pressure limited, volume control mode? Since it is presented under methods the reader immediately assumes it is somehow presenting characteristics of each test lung individually. Otherwise it is not clear how Sim1 “sees” a PIP pressure of 15 while the overall ventilator is set to a PIP of 35 or 40, unless these are actually “results” of using the resistance valve to adjust airflow (this is the case, presumably).

Response: Thank you very much for this observation. The location of the table in the manuscript was mistaken, but now it has been relocated to where it was supposed to be, the results section. Additionally, the legend was amended in aims to make it clearer for the reader. Since two tables were added, this is now called Table 4. 

1. Results: The data files uploaded into the OS database are, without exception, corrupt, unreadable, or appear to be empty CSV files when this reviewer tried to access them. Revision/reuploading is needed.

Response: We are not sure what happened during the upload process but we are sure that the files are correctly uploaded and will be available. Our apologies for this technical problem. 

1. Is table 1 presumed changes or were there actual measurements made? The table caption says “proposed” but the text P 12 L17 says “recorded”. If they were recorded, were they quantified and/or were the “recorders” blinded somehow? PEEP seems particularly difficult to estimate without a quantitative measurement, yet P17 L 238 states that PEEP could be set without impacting PEEP on the other lung unit – how were changes in PEEP determined? If the “eyeball” test was used throughout phase I, this should be stated. As written, it sounds as though they were measured, but the methods of measurement are not reported and qualitative results are presented.

Response: Thank you very much again. Table and text have been fixed

1. Was the short circuit tubing occluded for the 5 second VTe determinations? Does the short circuit tubing impact the accuracy of this method of measurement as a surrogate for what’s happening when the entire dual ventilation unit is in line on both lung units?

Response: Thanks for pointing this issue out. The flow through the bypass tubing has been discussed. 

1. P19 L 270-272 “the required adjustment on the ventilator setting (e.g. increased f) imposed changes (e.g. reduced VTe to maintain minute ventilation) on the other model.” Is unclear. Is f frequency (i.e. respiratory rate)? That is the first time this is mentioned in this document.

Response: You are absolutely right. Fixed up.

1. The animal model is an excellent addition, but with only two sets of animals, conclusions are extremely difficult to draw and statistical comparison is not presented (there appears to be a considerable amount of heterogeneity, so perhaps no comparisons demonstrated significant differences). There does appear to be a profound difference in the two “degrees” of ARDS in terms of P/F ratio, but no clear/consistent difference between them in the important variable most discussed in this manuscript, compliance. As such, the differential in applied PEEP is of interest, but the differences (or lack thereof) in other respiratory parameters (achieving an equivalent VTe, for example) is less exciting. Perhaps the manuscript could focus somewhat more on the manipulation of PEEP as the novel finding. Understandably, in the situation where ventilator splitting is needed, titration of both settings is necessary so both elements do need to be discussed.

Response: We agree. Please see our comment above.

1. Animals of different size + degree of ARDS may have given a more robust difference between compliance/tidal volume needs, and would have more dramatically illustrated the capability of this circuit to provide differential ventilation and maintain the ventilator parameters presented.

Response: Again! we agree. The study was very small and the sample did not manage to eliminate all doubts and questions. However, as a preliminary ´mini´-study, it opens the debate and generates proposals for new studies. 

1. Conclusions: The effect of the adjustable resistance on tidal volume will vary depending on respiratory rate/insp time; these effects must be discussed, stressing that resistance would need to be adjusted if ventilator respiratory rate or inspiratory time settings were changed, even if inspiratory pressure settings remained the same (also discussed in Cherry, et al). If the investigators could provide test data for varying respiratory rates with their setup, it would add significant novelty and interest.

Thank you for this very valuable suggestion. We added this data to our results and a brief discussion on what was observed. As with single ventilation, this further strengthens the fact that during dual ventilation, all changes in the ventilator will affect both paired units.

Reviewer #3: 

The authors have developed an interesting device and are commended on their work. The article is well written and does address some important aspects of differential multiventilation. However, there are some very important problems with the manuscript that must be addressed. The most important issues are that some of the conclusions are directly contradicted by works the authors reference as well as other available literature, the study has some critical limitations that limit some of their conclusions, and that the authors present their findings primarily in a qualitative manner and do not present quantitative results. Finally, while the authors link to their data, the files do not contain any data that could be used to validate their findings. Again, the authors have conducted some relevant and important work that would likely be of interest to the global community once these largely resolveable problems with the manuscript are addressed.

Response: Thank you very much for your nice comments, insights and suggestions to improve. We hope the new version of the manuscript fixes those flaws.

Major conceptual issues:

1. The authors link to their data, however the files do not contain any data that could be used to validate their findings and the OSF websites lists them as "Table empty or corrupt." The data should be reuploaded ensuring the character encoding of their comma-delimited files are compatible with the OSF site.

Response: We are not sure what happened during the upload process but we are sure that the files are correctly uploaded and will be available. Our apologies for this technical problem. 

The 3D models (.STL files) have been added to the repository. Link: https://github.com/ACRA2020/Pinch_Valve_design

The supplementary information (repository) is available on-line in a Dropbox link: https://www.dropbox.com/sh/nys8q88rupr3b3o/AAAog1Rq9iGrv97xW3GkAQc4a?dl=0

2. It is not clear that the shutoff valve in the short-circuit tubing is necessary and it's presence raises several concern. If the valve were accidentally closed, it would dramatically change the behavior of the device due to the absence of a short-circuit or bypass-circuit. The full implications are reviewed in by Roy et al. last year (Crit Care Explor, 2020, 10.1097/CCE.0000000000000198), but in brief, these could include:

-potentially ramping up the inspiratory pressure in both circuits,

-causing unpredictable behavior,

-triggering the obtruction alarm,

-or worst, causing no detectable problem immediately, caregivers leave the room with both patients alone in an isolation room with a device configuration that will suddenly causes the ventilator to behave erratically only once another problem occurs (e.g. single patient circuit obstruction).

Response: Thank you for your comment. Indeed, its clarification is more than relevant. Thank you also for sharing this information. Roy et al. manuscript clarifies these issues elegantly. 

We appreciate your criticism and take your suggestions for modifying the bypass circuit. We have eliminated the three-way stopcock from the device and replaced it by clamping when necessary, we have updated the figures. New measurements were made without the 3-way stopcock and Table 1 was modified.

However, the bypass tube or short circuit should be clamped in order to measure plateau pressure and the corresponding static compliance. We clarified this point in the text. Additionally, we decided to rename it as ‘bypass tubing’ in order to be in concordance with prior publications on this topic. 

1. Another problem with the short-circuit is that, because there is no one-way valve in the short circuit, expired CO2 or pathogens may potentially recircuilated through the circuit (particularly if respiratory effort is made) and be rebreathed by one or both patients. The authors did not test for this potentially crucial pitfall and very concerning possibility. Therefore, if the valve is left open during regular use, then this major potential issue must be discussed in the paper. If the valve is closed during regular use, then the previous issue becomes an even more major concern.

Thank you very much for your critical insight and the challenge that this proposed to us. We are aware that most literature on this topic uses a 22 mm by-pass circuit. However, we chose to use a 6 mm diameter and 15 cm length tube. This was selected taking into account previously obtained results from the numerical model described in (Eijo PMG, D’Adamo J, Bianchetti A, et al (2021) Exhalatory dynamic interactions between patients connected to a shared ventilation device. PLoS ONE 16, e0250672. doi: 10.1371/journal.pone.0250672), from our same research group. In order to select the size of the tube, two important factors must be taken into account: 1) The flow resistance of the by-pass circuit is proportional to the tube length and to the diameter elevated to the fourth power and, 2) the total air volume of the short-circuit tube is proportional to the length and to the Diameter^2. In our study, the selected tubing guaranteed a negligible flow from expiration to inspiration limbs. The integration of this flow along time results in the total air volume associated with this flow (2.9 mL), which is lower than the tube volume (5.6 mL). In our numerical tests, the possibility of cross contamination between patients associated with this flow was almost null. Please refer to the figure below. 

Figure: Left) Ventilator pressure vs time, Middle) Pressure Difference between the ends of the tubing during time, Right) Flow rate through the tube against time (positive values from inspiration limb to expiration limb) 

We understand that the use of a one-way valve would have been beneficial, and therefore we will take this into consideration for future developments of the ACRA device. This was also added to the text. 

1. On line 330, the authors state that the "simultaneous use of two PEEP valves, however, can result in inadequate pressure signals that lead to the ventilator not reaching its target PEEP, alarm triggering and undesirable ventilator responses and would be, therefore, not recommended." 

There are two problems with this statement: First, it is not technically correct. The predictable behavior of the ventilator depends on the ventilators ability to self sense. The problems described by the authors are seen when there is no adequate bypass circuit AND there is modification of either inspiratory or expiratory pressures in BOTH circuits. If the valve on their device's short-circuit is closed and/or the tubing provides inadequate flow (i.e. due to the small tubing with a narrow opening through the valve), the authors will likely still see such behavior when the PEEP valve is set to higher settings and the pinch valve of the other circuit is set sufficiently closed. These implications are described in detail in the Roy et al. paper (Crit Care Explor, 2020, 10.1097/CCE.0000000000000198). Moreover, the second problem with the statement on line 330 is that the authors have, in their manuscript, cited Raredon et al. who provided a very thorough testing of device with exactly the design the authors claim to be problematic. Not only did Raredon et al not see this behavior, but they directly contradict the authors statement and explicitly describe the opposite. At very least, the authors must address in the discussion section how such a major discrepancy could exist between their findings and the contradictory findings of Raredon et al.

Response: Thank you very much for clarifying this point. Indeed, our phrase presented conflicts. It has been rephrased and the vision of Roy et al. and Raredon et al. have been discussed.

1. Table 1 provides a qualitative overview of what should happen with use of the author's device. However, there are two problems in the manuscript that are particularly noticed in this table: first, the table is presented as a table of results, in which case quantitive rather than qualitative results are appropriate. As in other parts of the paper, the authors report qualitatively summarized results when detailed quantitative data would be more appropriate. The second, perhaps larger issue is the methodological: the loosely described protocol appears to involve manipulation of only one variable at a time. Manipulation of only one variable at a time does not allow for the interaction between components. As outlined above, there are major possible (even likely!) interactions that could occur here and they have been described in detail in the literature. In fact, such interactions would have been very likely occur in the author's device during testing since the short circuit was occluded (and perhaps, even if it had not). The authors must provide supporting data for their conclusion, address how their protocol (including all of the tests) would detect interactions between components, and discuss any remaining limitations in the protocol.

Response: Thanks for your comments. The table has been modified and the results expressed quantitatively. We hope it is clearer now.

We agree that there are potential interactions that have been contemplated by the measurements. The objective of phase 1 was to define whether there was independence between the units ventilated to action by changes on the opposite side and the ventilator. The results clearly demonstrate that changes in the ventilator have an impact on both terminals and that individual modifications, although extreme, only affect the driven side.

We add new results. We hope you will find them interesting.

1. Table 2 demonstrates that only a fixed PIP, a fixed PEEP, a fixed respiratory rate and a fixed I:E ratio were used for testing. Because a pinch valve is used in the inspiratory circuit rather than a pressure release valve (as others have used), the pressure drop across the pinch valve will change based on the Inspiratory time, the ventilator set PIP & PEEP, and the setting on the PEEP valve. This major caveat results in non-linear changes in the PIP when any of these 3 settings are changed. By using only a single fixed ventilator setting for the testing, none of the effects that result from this caveat are able to be observed. In real use, it is likely that both pinch valves may need adjusted anytime a change is made to the I:E ratio, the I time, the PIP, or the PEEP. These important non-linear effects of ventilator setting changes is not seen in the authors' testing, but remains highly relevant to the use of the device. Ideally, the testing protocol would be more developed and address these issues. At minimum, these caveats and their potential repercussions must be discussed in the appropriate section.

Response: Thank you for your comments. We added new information to assess the changes in the VTe when changing respiratory rate and inspiratory time.

Other issues

1. abstract: The abstract does not include any of the results. Similar to the body of the manuscript, the authors articulate success (e.g. "feasible") rather than results ("consistent","stable","unaltered", etc). Please add a sentence that summarizes the measurement results i.e. what outcome measure(s) indicated that it was feasible.

1. line 23: The authors states the components are all "approved", but it is not clear who has approved them and is doubtful that all were approved for use in this manner. Approval is given to medical devices for specific uses. For example, many countries require separate certification for materials that are in the breathing pathway (e.g. ISO 18562). Are the parts all approved for use within the gas path according to such a specification? Who approved them for such use?

Response: Thank you for your comments. The text was amended. 

1. line 25/figure 2: The PEEP valve is place on the expiratory limb of the unit that would require the greater PEEP. However, what happens if the individual who requires the greater PEEP changes? The authors mention in the discussion that this might occur, but do not address how it would be fixed... Is it possible to switch over the valve or does this require disassembly?

Response: Thank you for your comments. The text was amended. 

1. line 330: the authors cite 3 papers to support the notion that inline PEEP valve may be used to modify the PEEP. The cited article by Clarke et al. does not support this statement as it does not discuss inline PEEP valves at all. Additionally, the author's device appears to utilize a particular type of inline modification of a commercial PEEP valve described by Bunting et al. (AJEM, 2020, 10.1016/j.ajem.2020.06.089) which was not cited.

Response: Thank you for your comments. The text was amended

1. line 370 & 390: VTe can be individualized but only indirectly through changes in pressure. It should be stated clearly that changes in Vte are can only be individualized by means of pressure adjustment and use of such a device in a Volume Control mode would be dangerous.

Response: Thank you for your comments. The text was amended. 

Reviewer #4: 

This paper by Otero et al. discusses an interesting new device to enable a more accurate control of shared ventilation. Shared ventilation is mostly discouraged by most professional societies (as also discussed by the authors) permitting it only in specific cases and while ventilator shortage does not seem to be as a major issue as it was at the beginning of the pandemic, the concept of ventilation sharing can still be applied in emergency settings and extra means to promote the safety of this practice is welcome in my opinion.

Response: Thank you very much for your insights and suggestions. We hope the new version of the manuscript will fix those flaws.

The 3D models (.STL files) have been added to the repository. Link: https://github.com/ACRA2020/Pinch_Valve_design

The supplementary information (repository) is available on-line in a Dropbox link: https://www.dropbox.com/sh/nys8q88rupr3b3o/AAAog1Rq9iGrv97xW3GkAQc4a?dl=0

My questions and opinions are the following:

1. Did the authors record or assess the flow patterns of the individual circuits?

Response: Yes, flow patterns were recorded in every phase for each of the paired unit. However, the evaluation of this data, although undoubtedly valuable in a real case scenario, was not part of the aims of this study, and the management of the dual ventilation was guided based on pressure measurements. The lack of analysis of these curves is stated in limitations. Due to the importance of this topic in the field of dual ventilation, this has been studied in detail in another publication from our study group (https://doi.org/10.1371/ journal.pone.0250672). 

2. This device seems to manipulate the resistance of the individual circuits by the pinch valves to compensate for the differences in compliance, resulting in impedances with a suitable ratio to deliver appropriate tidal volumes. In this regard, the principle behind this device seems to be somewhat similar to another study recently published (doi: 10.1016/j.resp.2020.103611). This should be discussed in the manuscript.

Response: Thank you for this observation and for the shared reference. We absolutely agree with this concept. Indeed, our research group has recently published a study in this regard (https://doi.org/10.1371/ journal.pone.0250672). In order to comply with your request, we added a comment about this in the discussion. We hope you agree with this addition. 

3. The use of manual pinch valves can be problematic if patient dynamics change, since it seems that the system cannot deliver alerts in case ventilation to the specific patients change. This can mean an even higher burden on the ICU staff already under high pressure to actively monitor the manometers, in my opinion actually restricting the use of the device more than the need for ventilators in most ICUs. The single extra PEEP valve can also pose some risks: as the authors describe it should be place in the circuit which seems to require a higher PEEP. However, the course of the disease can be quite different even in patients that are similar at the time of recruitment, resulting in a change of “higher PEEP circuit”, requiring the complete swap of the circuits and re-adjusting all the pinch valves to deliver appropriate volumes.

Response: Yes, we agree that the device proposed has limitations, many of which are common to the process of shared ventilation itself. We believe that this method of shared ventilation, as well as all the other proposed, are not intended to provide optimal conditions for ventilation, but to be used as the last resource when there is no better option for a patient in respiratory distress. 

Changes of the delivered tidal volume to any of the patients or on the compliance or resistance of each circuit are able to be detected by the alarm system of the ventilator. The proposed device is only an interphase and has no alarm system, but the alarms of the ventilator still play a major role. In fact, the alarms of the ventilator should be set prior to the setting up in order to detect any alterations (as described by Beitler JR, Kallet R, Kacmarek R, et al Ventilator sharing protocol: dual-patient ventilation with a single mechanical ventilator for use during critical ventilator shortages). This was acknowledged in the discussion. 

Regarding the variation of the necessity of the PEEP valve as disease progresses, we agree that this is a limitation of the current design of the device, and this has been stated in the manuscript. We believe that this device is not intended to replace a ventilator but only to increase its capacity, by preferably using it in relatively stable patients and for a short period of time. Perhaps the interposing of two PEEP valves (one for each unit) could be a possible solution as long as only one of them is used at the same time. In this case, the practitioner would have the possibility to choose which valve to use. We have not studied this and therefore it only remains a hypothesis for further experiments. A proposed solution is also mentioned in the text. 

4) In Table 1 the authors describe that increasing PEEP in Lung 2 results in an increased PEEP in Lung 1 and no change in Lung 2. I presume this is a typo and the authors meant to describe the effects of a PEEP increase on Lung 1. While having constant settings on the ventilator, setting the pinch valve on one lung should also have an effect on the ventilation of the other circuit, since the ratio of impedances between the two circuits is going to be altered, resulting in a different split of delivered volume. However, the authors describe no effects on the other circuit (Table 1, Pinch valve on Lung 1 and 2).

Response: Thank you very much for your attentive revision of this table. Indeed, this was a typo and we are so glad this was detected. However, per request of another reviewer, this table was modified (with numerical values) and this was also amended.

5) I don’t really see how the occlusion of one circuit to measure the volume won’t affect the delivered volume to the other end, since the practically infinitely high resistance of the blocked ET tube should reroute some of the volume to the open lung, since the total compliance of the circuit should be different. Did the authors directly measure each tidal volume by placing flow meters into the individual circuits to assess this or did they assess the individual tidal volumes only using this method? I have also similar doubts about the lack of effect on the other patient in case of a disconnection, since the open tube of the disconnected patient should also change the flow distribution. If the valve of the patients is closed (in case of a planned disconnection), then the ventilator should be somewhat adjusted for the changed total compliance of the circuit.

Response: Yes, indeed. Unlike during volume controlled ventilation, during pressure controlled ventilation the occlusion of one circuit will not elicit any modification of the pressure or volume of the paired unit. This was corroborated during the experiments and the additional information (videos) was added to the repository.

Regarding the case of a disconnection, we agree that this concept may elicit doubts. We have noted that not all ventilators can compensate for this loss. In our experiments, this was compensated and corroborated (additional data was uploaded). 

6) While the use of the short circuit tubing is a viable method to trick the ventilator to using different amounts of PEEP, it also carries some risks due to the lack of some alerts and also since this short circuit is actually a low-impedance third patient that would also need to be balanced according to the actual clinical scenarios in the two real patients. This would necessitate a third flow meter to verify that the short circuit tubing is not stealing considerable amounts of air from the real patients.

Response: the functions and importance of the bypass tubing (our short tubing) is amended in the text. 

7) There seems to be a typo on Fig. 2: the captions states that #6 is the PEEP valve and it seems to be the case based on the photo as well. However, while the caption states that #7 and #7’ are the connections for the expiratory tubing and #8 and #8’ are the connections for the inspiratory tubing, based on the pictures this should be the opposite in my opinion, with #7 to be the inspiratory and #8 to be the expiratory port.

Response: Thank you very much for this observation. This was a mistake on the numbers on the picture which we have already amended. 

8) What kind of ICU ventilators did the authors use? Did they face any alerts from the ventilator due to the use of the splitter device?

Response: In this study we used a Nellcor 65 Puritan Bennet 760. The use of this ventilator with the shared device did not set off any alarm, and delivered a tidal volume according to the preset variables. Alarms on the ventilator functioned normally, as if only one patient was connected.

---

## [Editor Report · Decision Letter 1]

9 Aug 2021

Ventilator output splitting interface 'ACRA': description and evaluation in lung simulators and in an experimental ARDS animal model

PONE-D-21-13259R1

Dear Dr. Otero,

We’re pleased to inform you that your manuscript has been judged scientifically suitable for publication and will be formally accepted for publication once it meets all outstanding technical requirements.

Kind regards,

Aleksandar R. Zivkovic

Academic Editor

PLOS ONE

---

## [Editor Report · Acceptance letter]

16 Aug 2021

PONE-D-21-13259R1 

Ventilator output splitting interface 'ACRA': description and evaluation in lung simulators and in an experimental ARDS animal model 

Dear Dr. Otero:

I'm pleased to inform you that your manuscript has been deemed suitable for publication in PLOS ONE. Congratulations! Your manuscript is now with our production department. 

Kind regards, 

on behalf of

Dr. Aleksandar R. Zivkovic 

Academic Editor

PLOS ONE